Corrected: Author correction

# Rad52 prevents excessive replication fork reversal and protects from nascent strand degradation

Eva Malacaria[1], Giusj Monia Pugliese[1], Masayoshi Honda[2], Veronica Marabitti[1], Francesca Antonella Aiello[1], Maria Spies[2], Annapaola Franchitto[1] & Pietro Pichierri [1,3]

Stabilisation of stalled replication forks prevents excessive fork reversal and their pathological degradation, which can undermine genome integrity. Here we investigate a physiological role of RAD52 at stalled replication forks by using human cell models depleted of RAD52, a specific small-molecule inhibitor of the RAD52-ssDNA interaction, in vitro and single-molecule analyses. We demonstrate that RAD52 prevents excessive degradation of reversed replication forks by MRE11. Mechanistically, RAD52 binds to the stalled replication fork, promotes its occlusion and counteracts loading of SMARCAL1 in vitro and in vivo. Loss of the RAD52 function results in a slightly-defective replication restart, persistence of under-replicated regions and chromosome instability. Moreover, the RAD52-inhibited cells rely on RAD51 for completion of replication and viability upon replication arrest. Collectively, our data suggest an unexpected gatekeeper mechanism by which RAD52 limits excessive remodelling of stalled replication forks, thus indirectly assisting RAD51 and BRCA2 in protecting forks from unscheduled degradation and preventing genome instability.

[1] Mechanisms, Biomarkers and Models Unit, Department of Environment and Health, Istituto Superiore di Sanità, Viale Regina Elena 299, 00161 Rome, Italy. [2] Department of Biochemistry, Carver College of Medicine, University of Iowa, 51 Newton Road 4-403 Bowen Science Building, Iowa City, IA 52242, USA. [3] Istituto Nazionale Biostrutture e Biosistemi, Viale delle Medaglie d'Oro, 305, 00136 Rome, Italy. These authors contributed equally: Giusj Monia Pugliese, Masayoshi Honda.  Correspondence and requests for materials should be addressed to P.P. (email: pietro.pichierri@iss.it)

Replication stress is one of the most important drivers of genome instability in normal and cancer cells. Hence, cells have evolved multiple mechanisms to prevent incorrect handling of perturbed replication forks. Recently, processing and remodelling of perturbed replication forks, and especially the reversal of the stalled forks, emerged as critical events in the correct recovery of replication and in the maintenance of genome stability. Replication fork reversal (RFR) involves the regression of the fork accompanied by the subsequent reannealing of the two extruded nascent strands, resulting in the formation of a Holliday junction (HJ)-like structure[1]. Several motor proteins can promote RFR in vitro and some of them, such as SMARCAL1, HTLF, ZRANB3 and FBH1, are associated with RFR also in the cell[2–4]. The reversed fork (RF) provides an intermediate for subsequent replication restoration by multiple means, including recombination. However, its double-strand break (DSB)-like end makes the RF vulnerable to nucleolytic attack[2,3,5]. While a controlled end-processing at the extruded arm of a RF seems instrumental to fork restoration, an unscheduled nucleolytic digestion is actively prevented[2]. This RF protection is largely achieved by BRCA2-dependent coating of nascent strand with RAD51 recombinase in a BRCA2-dependent mechanism[6–8]. Cells depleted of BRCA2 are a paradigm of fork protection mutants in which RFs are extensively degraded by MRE11 in combination with additional nucleases[6,7,9–12]. Such pathological degradation of the RF is correlated to the response to chemotherapy of BRCA-deficient cancer cells and may drive genome instability[10,13–15].

Although recent findings explained how RAD51, BRCA1/2 and other factors contribute to RF stabilisation, it is unknown how stalled forks are channelled to RFR, since it was demonstrated that excessive RFR is as detrimental as its absence[3–5,15,16].

In yeast, Rad52 is a recombination mediator that promotes homologous recombination (HR) by facilitating the exchange of single-stranded DNA (ssDNA) binding protein Replication Protein-A (RPA) with Rad51 recombinase on ssDNA[17–19]; a function that in human cells is carried out by BRCA2[20–22]. Recently, human RAD52 has been shown to participate in the recovery of collapsed replication forks by break-induced replication (BIR) and in the replication stress-related mitotic DNA synthesis (MIDAS)[23,24]. Thus, RAD52 may be involved in the rescue of perturbed replication forks under pathological conditions. Whether it contributes to the integrity of stalled forks in normal cells remained unknown. RAD52 possesses features that may be employed at a stalled fork. Its ability to multimerize and to interact with ssDNA, double-stranded DNA (dsDNA) and RPA[25] may be useful in preventing excessive loading of fork reversal enzymes at the exposed parental ssDNA formed once the fork gets blocked. Furthermore, RAD52 is recruited to nuclear foci after replication fork arrest[26].

Here, using RAD52-depleted human cells and a small-molecule inhibitor that abrogates the RAD52–ssDNA interaction[27], we investigated the role of RAD52 in the stabilisation of stalled replication forks. We report that abrogation of RAD52–ssDNA binding results in extensive nascent strand degradation by MRE11 after fork reversal. We find that RAD52 can bind to stalled forks in vivo and stimulates the association between the two arms of the replication fork in vitro. Notably, we find that RAD52 prevents super-physiological recruitment of fork reversal enzymes, including SMARCAL1, after replication fork arrest in vivo and antagonises SMARCAL1-mediated fork reversal in vitro. This new function of RAD52 is essential for a correct recovery from the replication arrest as RAD52 inhibition results in under-replication and a genuine reliance on RAD51 for viability.

Altogether, our findings unveil a previously unidentified function of RAD52 in fork protection and recovery, which may be critical for genome integrity of normal cells and for the observed BRCA2–RAD52 synthetic lethal relationship.

## Results

**RAD52 protects stalled replication forks from degradation.** To study how RAD52 contributes to replication fork stability, we generated MRC5SV40 cells stably depleted of RAD52 by RNA interference (RNAi). As shown in Fig. 1a, transduction with two different short hairpin RNA (shRNA) lentiviruses ($V_1$ or $V_2$) resulted in little residual RAD52 as compared with control-infected cells ($V_c$–shCTRL). Cells infected with the $V_2$ lentivirus are referred to as shRAD52 throughout our study. Accumulation of ssDNA in the nascent strand is an acknowledged readout of stalled replication fork instability and degradation[3,28]. Thus, we performed nascent ssDNA detection by native iododeoxyuridine (IdU) immunofluorescence assay in shCTRL, shRAD52 cells or in the parental MRC5SV40 cells treated with the small-molecule RAD52 inhibitor epigallocatechin (EGC, RAD52i), which recapitulates the phenotype of shRAD52 cells[27]. In control cells, nascent ssDNA was barely detectable during unperturbed replication, while it was significantly elevated by hydroxyurea (HU)-induced replication fork arrest (Fig. 1b). Compared to control cells (shCTRL), depletion or inhibition of RAD52 greatly increased early exposure of nascent ssDNA in HU, but resulted in a reduction at 6 h of treatment (Fig. 1b). Add-back of the RNAi-resistant form of RAD52 in the shRAD52 cells restored a wild-type level of nascent ssDNA in HU-treated cells (Fig. 1c, d). To confirm that exposure of nascent ssDNA was confined to S-phase, we labelled replicating cells with EdU 10 min before addition of IdU and sampling. In unperturbed cells, IdU-labelled ssDNA was almost completely confined to EdU-labelled S-phase cells (Supplementary Fig. 1a). Anti-RPA32 immunofluorescence in detergent-extracted cells also revealed increased formation of ssDNA (Supplementary Fig. 1b), which was also recapitulated by transduction with the other shRAD52 virus ($V_1$), confirming the phenotype (Supplementary Fig. 2a). To determine if accumulation of nascent ssDNA correlated with end-resection of DSBs at stalled replication forks, we performed neutral Comet assay. As previously reported[29], in wild-type cells (dimethylsulphoxide (DMSO)), DSBs were not substantially induced by HU up to 4 h of treatment, as shown by the round-shaped nuclei (Fig. 1e). DSBs started to be clearly detectable only beyond 4 h of treatment and were abundant at 24 h (Fig. 1e). Inhibition or depletion of RAD52 did not affect formation of DSBs at early time points of treatment, while it significantly reduced their amount at 6 h (Fig. 1e and Supplementary Fig. 2b), suggesting that the observed reduction of nascent ssDNA at this time point was linked to RAD52-dependent formation of DSBs, as previously reported[27].

Exposure of ssDNA has been correlated with fork degradation by MRE11-EXO1 or DNA2[2,3]. To test whether accumulation of nascent ssDNA was dependent on MRE11 in the absence of RAD52, we exposed cells to the MRE11 inhibitor MIRIN prior to labelling the nascent DNA. Inhibition of MRE11 restored wild-type levels of nascent ssDNA in RAD52-inhibited or shRAD52 cells, under unperturbed conditions and upon replication arrest (Fig. 1f and Supplementary Fig. 2c).

Processing of stalled forks initiated by MRE11 can lead to extensive degradation of nascent strand[10,11], which can be visualised also as the shortening of replication tracks[30]. Thus, we labelled ongoing replication forks by a dual clorodeoxyuridine (CldU)-IdU pulse before treating cells with HU and the RAD52i, alone or in the presence of MIRIN (Supplementary Fig. 3a). As expected, analysis of the IdU tracts in dual-labelled replication tracks showed that little nascent strand degradation occurs after 4 h of HU in wild type, and that this reduction was

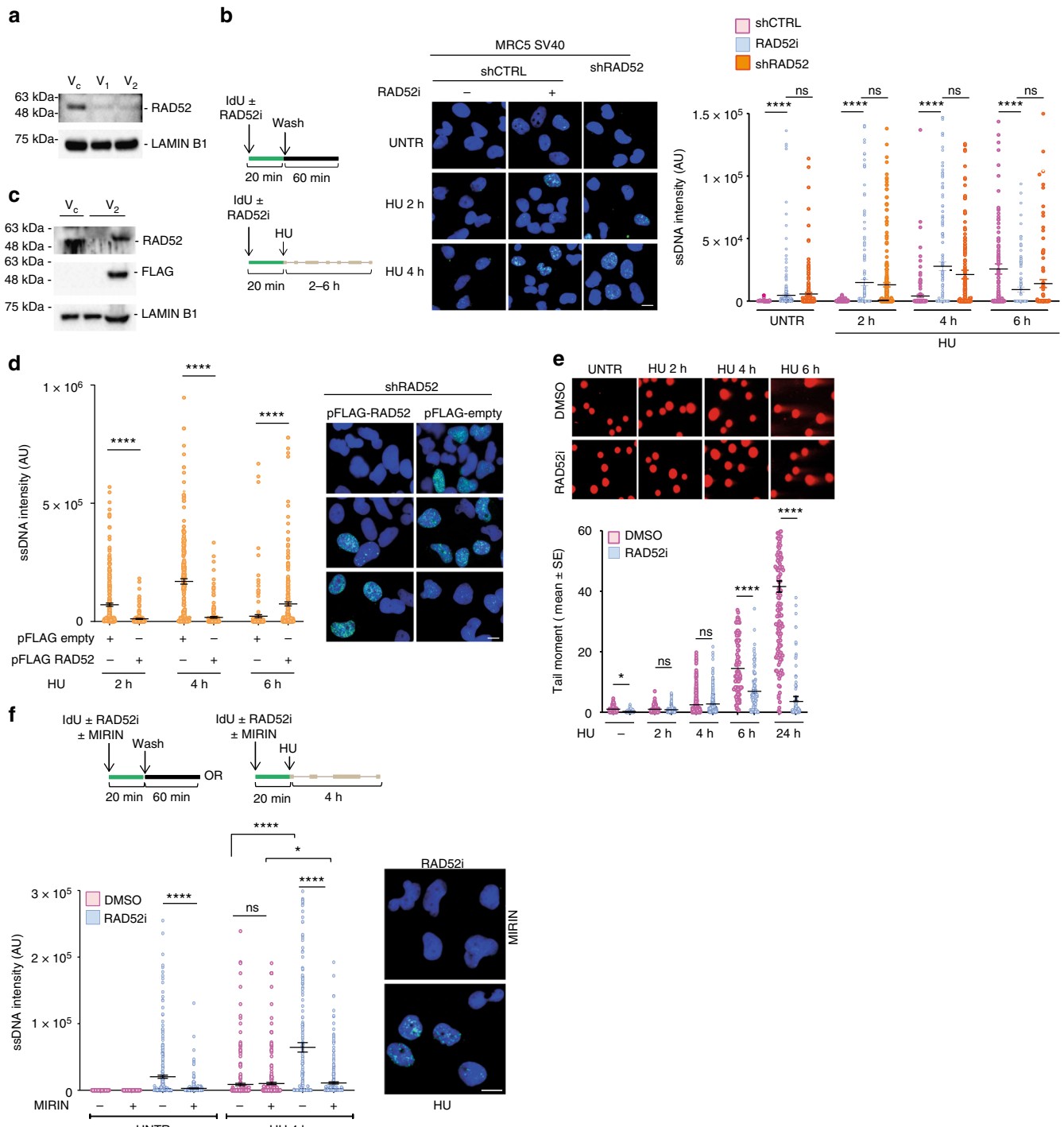

**Fig. 1** Loss of RAD52 causes MRE11-dependent accumulation of nascent ssDNA. **a** Western blot shows level of RAD52 protein in cells were infected with two different viruses ($V_1$ or $V_2$) containing short hairpin RNA (shRNA) sequences against RAD52. LAMIN B1 was used as loading control. C = control virus. **b** Analysis of nascent ssDNA after 2 mM hydroxyurea (HU) treatment. On top: schematic of the experiment. Graph shows the intensity of ssDNA staining for single nuclei. Values are presented as means ± SE (ns = not significant; ****$P < 0.0001$; Mann–Whitney test; $N = 184$). Representative images are shown. **c** Western blot analysis of FLAG-RAD52 transfection. LAMIN B1 was used as a loading control. **d** Cells were transfected with FLAG-RAD52 or FLAG-empty 48 h before to perform nascent ssDNA immunostaining. Dot plots show the mean intensity of ssDNA staining for single nuclei from each cell line after treatment with HU 2 mM. Values are presented as means ± SE (****$P < 0.0001$; Mann–Whitney test; $N = 184$). Representative images are shown. **e** Cells were treated with 2 mM HU at different time points and collected to perform neutral comet assay. Data are presented as mean tail moment ± SE from two independent experiments. $N = 93$ (ns = not significant; *$P < 0.1$; ****$P < 0.0001$; Mann–Whitney test). Representative images are shown. **f** Cells were treated as indicated in the schemes. Graph shows the mean intensity of ssDNA staining for single nuclei from each cell line. The intensity of the anti-iododeoxyuridine (IdU) immunofluorescence was measured in at least 100 nuclei from two independent experiments. Values are presented as means ± SE (ns = not significant; *$P < 0.1$; ****$P < 0.0001$; Mann–Whitney test; $N = 290$). Representative images of ssDNA formation are given. Scale bar represents 5 μm. Source data are provided as a Source Data file

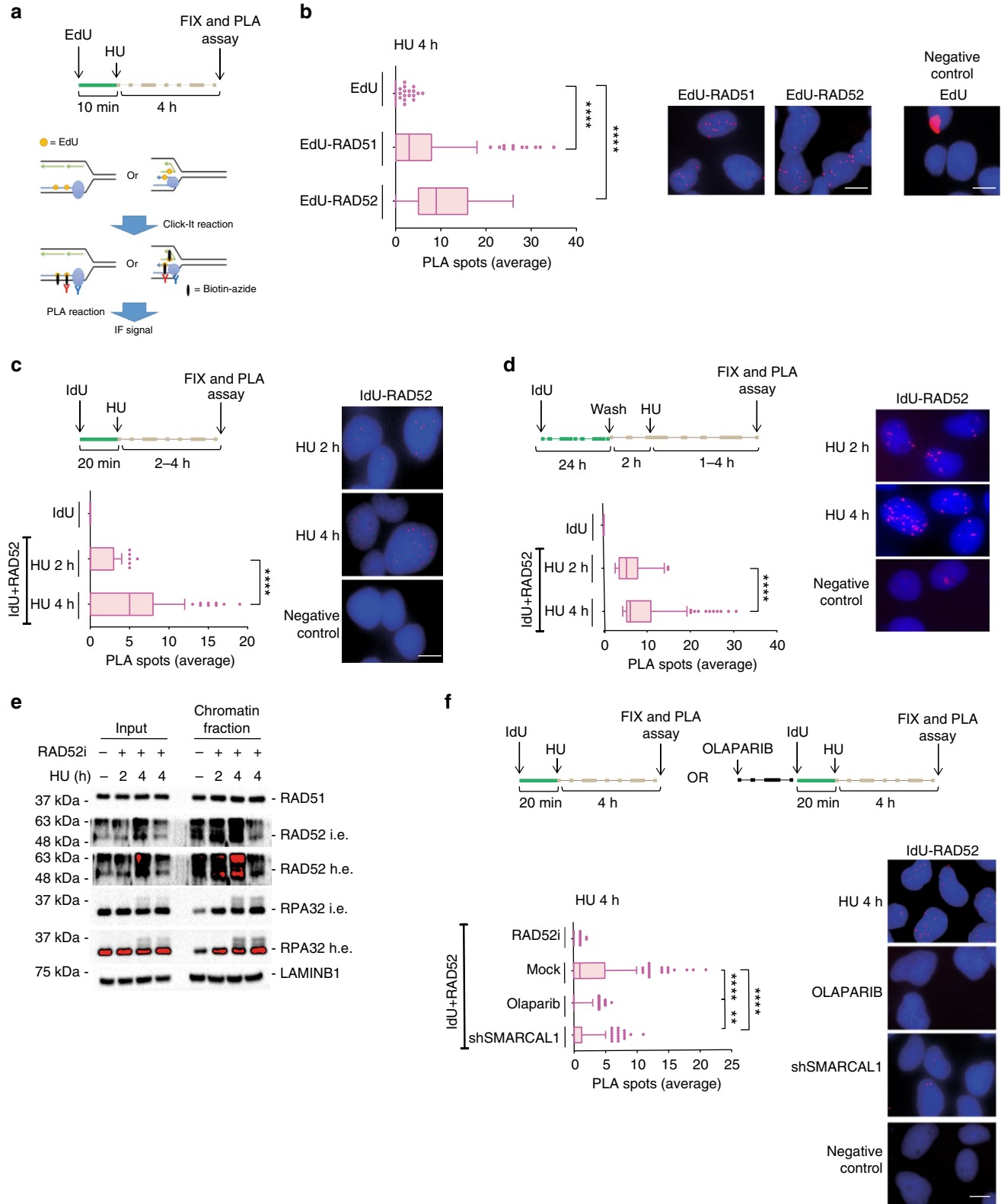

MRE11 independent (Supplementary Fig. 3b). Although RAD52 inhibition resulted in accumulation of nascent ssDNA after HU, the analysis of DNA fibres did not show any significant reduction in the IdU tracts, which were surprisingly reduced rather than increased by MIRIN treatment (Supplementary Fig. 3b). However, treatment with S1 nuclease to degrade any ssDNA prior to immunodetection of the replication tracks (Supplementary Fig. 4a) revealed shorter IdU tracts in RAD52-inhibited cells on HU, as compared with the non-inhibited cells (Supplementary Fig. 4b). Most importantly, MIRIN recovered this reduction in the IdU tract length in RAD52-inhibited cells.

**Fig. 2** RAD52 is recruited at perturbed replication forks. **a** Experimental scheme of in situ fork recruitment assay by EdU-proximity ligation assay (PLA). PLA was performed using anti-biotin and anti-RAD52(1) antibodies. Controls were subjected to PLA with anti-biotin only (EdU). **b** The graph shows the mean number of PLA spots per cell. Values are presented as means ± SE (****$P < 0.0001$; Mann–Whitney test; $N = 149$). Representative images are shown. **c** Analysis of nascent ssDNA–RAD52 interaction by native iododeoxyuridine (IdU)-PLA. On top, schematic of the assay. The graph shows the mean number of PLA spots per cell. PLA performed with anti-IdU only served as negative control. Values are presented as means ± SE (****$P < 0.0001$; Mann–Whitney test; $N = 135$). Representative images are shown. **d** Analysis of parental ssDNA-RAD52 by native IdU-PLA. DNA was labelled as described in the scheme. Graph shown the mean of PLA spots per cell ± SE. PLA performed with anti-IdU only served as negative control. Values are presented as means ± SE (****$P < 0.0001$; Mann–Whitney test; $N = 134$). Representative images are shown. **e** Analysis of chromatin recruitment of RAD52 and RPA32 after replication arrest. Input represents 10% of cell suspension lysed before fractionation. LAMIN B1 was used as a loading control. L.e. = short exposure; H.e. = long exposure. **f** Analysis of DNA–protein interactions by in situ PLA assay. Cells were treated as in the scheme. Graph shown the mean of PLA spots per cell ± SE. As PLA performed with IdU only—negative control—did not show spots it was not included in the graph, but it is provided as representative image. Values are presented as means ± SE (**$P < 0.01$; ****$P < 0.0001$; Mann–Whitney test; $N = 257$). Representative images are shown. Scale bar represents 5 µm. Source data are provided as a Source Data file

BRCA2 is the main mediator of RAD51 in humans[31], however, it has been speculated that human RAD52 can perform recombination mediator function under some conditions[32]. Treatment of cells with the RAD51 inhibitor B02[33] before HU did not increase exposure of nascent ssDNA at arrested replication forks, but rather led to its reduction (Supplementary Fig. 5a), which is consistent with observation that RAD51 depletion counteract RF formation and degradation[7,34]. Thus, the divergent phenotypes of RAD51 and RAD52-inhibited cells suggest that RAD52-dependent fork protection is unrelated to a RAD51 mediator function. Similarly, accumulation of nascent ssDNA after RAD52 inhibition or depletion was not recapitulated by MUS81 depletion (Supplementary Fig. 5b and c), indicating that the RAD52-MUS81 collaboration reported under pathological conditions[7,23,35] is not involved in fork protection in wild-type cells.

These results indicate that RAD52 protects nascent strand from MRE11-dependent degradation and that the fork protection function of RAD52 is independent of the role performed with the MUS81 at collapsed replication forks.

**RAD52 is recruited to ssDNA after replication fork arrest**. In human cells, conditions leading to fork collapse and DSBs induce recruitment of RAD52 to perturbed replication forks[23,36]. Our data demonstrate that RAD52 protects replication forks from degradation by MRE11 independently of formation of DSBs at the fork. Thus, we investigated if RAD52 was recruited to perturbed forks early after replication arrest when no DSBs are detected. Recruitment of RAD52 to forks was assessed at the single-cell level by 5-ethynyl-2'-deoxyuridine (EdU)-proximity ligation assay (PLA)[37], modified with a low-salt treatment to remove loosely bound factors. EdU-PLA evaluated the proximity of antibody-detected RAD52 with EdU-labelled nascent DNA (Fig. 2a). As a control, we performed EdU-PLA with RAD51, a protein found at stalled fork by multiple approaches. PLA detected a substantial interaction between nascent DNA and RAD52 after HU treatment (Fig. 2b). Of note, the number of EdU-RAD52 PLA signals, which reflects the level of recruitment at fork, was comparable with that of EdU-RAD51 (Fig. 2b).

EGC (RAD52i) interferes with RAD52 binding to ssDNA[27], suggesting that RAD52–ssDNA interaction may be essential for fork protection. After fork stalling, regions of ssDNA can form at the parental strand and/or at paired nascent strands after fork reversal. Because EdU-PLA does not discriminate which strand RAD52 associates with, we performed native IdU-RAD52 PLA, which can detect association of proteins specifically with nascent or parental ssDNA depending on the labelling scheme[30,38]. Although PLA did not detect association of RAD52 with nascent ssDNA under unperturbed conditions, IdU-RAD52 PLA spots were readily observed after replication arrest and significantly increased over time (Fig. 2c). Treatment with HU also stimulated the interaction of RAD52 with parental ssDNA and this interaction appeared more abundant or easily detected than that with nascent ssDNA (Fig. 2d).

Recruitment of RAD52 was also investigated by western blotting on chromatin after cellular fractionation (Fig. 2e). In agreement with single-cell data, treatment with HU increased the amount of RAD52 in chromatin, which was reduced after treatment with EGC, indicating that it derived from binding to ssDNA.

While exposure of ssDNA at parental strand mostly derives from helicase and polymerase uncoupling after HU treatment, detection of nascent ssDNA is linked to fork reversal[6]. Thus, we investigated whether nascent ssDNA-RAD52 recruitment occurred at RFs or downstream their formation. If this is true, any treatment minimising fork reversal would reduce RAD52 recruitment at nascent ssDNA. To interfere with fork reversal, we treated cells with the PARP1 inhibitor Olaparib, which prevents formation of RFs if used before fork arrest[39]. As expected, treatment with HU for 4 h led to association of RAD52 with nascent ssDNA and the RAD52i abrogated formation of ssDNA-PLA spots (Fig. 2f). Treatment with Olaparib also strongly reduced the association of RAD52 with nascent ssDNA (Fig. 2f), suggesting that most of this interaction occurs downstream fork reversal. Since SMARCAL1 is critical for fork reversal[6,12], we analysed interaction of RAD52 at nascent ssDNA after fork arrest by IdU-PLA in cells depleted of SMARCAL1. Consistent with the effect of Olaparib, knock-down of SMARCAL1 substantially reduced the interaction of RAD52 with nascent ssDNA at stalled forks (Fig. 2f). In contrast, association of RAD52 with parental ssDNA was not reduced in the absence of SMARCAL1 (Supplementary Fig. 6).

Collectively, our data indicate that replication fork stalling stimulates rapid recruitment of RAD52 to the fork where it associates with both parental and nascent ssDNA, and that interaction with nascent ssDNA requires the fork reversal factor SMARCAL1.

**Fork degradation occurs after fork reversal in RAD52i cells**. We show that RAD52 associates with nascent ssDNA downstream fork reversal (Fig. 2f). Exposure of ssDNA at nascent strand is an indirect readout of RFR[6,30,40]. Thus, we asked whether nascent ssDNA accumulating in the absence of RAD52 would derive from destabilised RFs. We interfered with fork reversal by exposing cells to Olaparib before fork arrest and analysed the presence of nascent ssDNA. Exposure of nascent ssDNA was greatly stimulated by RAD52 inhibition or depletion and was completely abrogated by pre-treatment with Olaparib (Fig. 3a and Supplementary Fig. 7). Suppression of the RAD52i phenotype by Olaparib suggests that accumulation of nascent ssDNA derives from extensive degradation of at least one of the two paired nascent strands of a RF. We therefore expected that

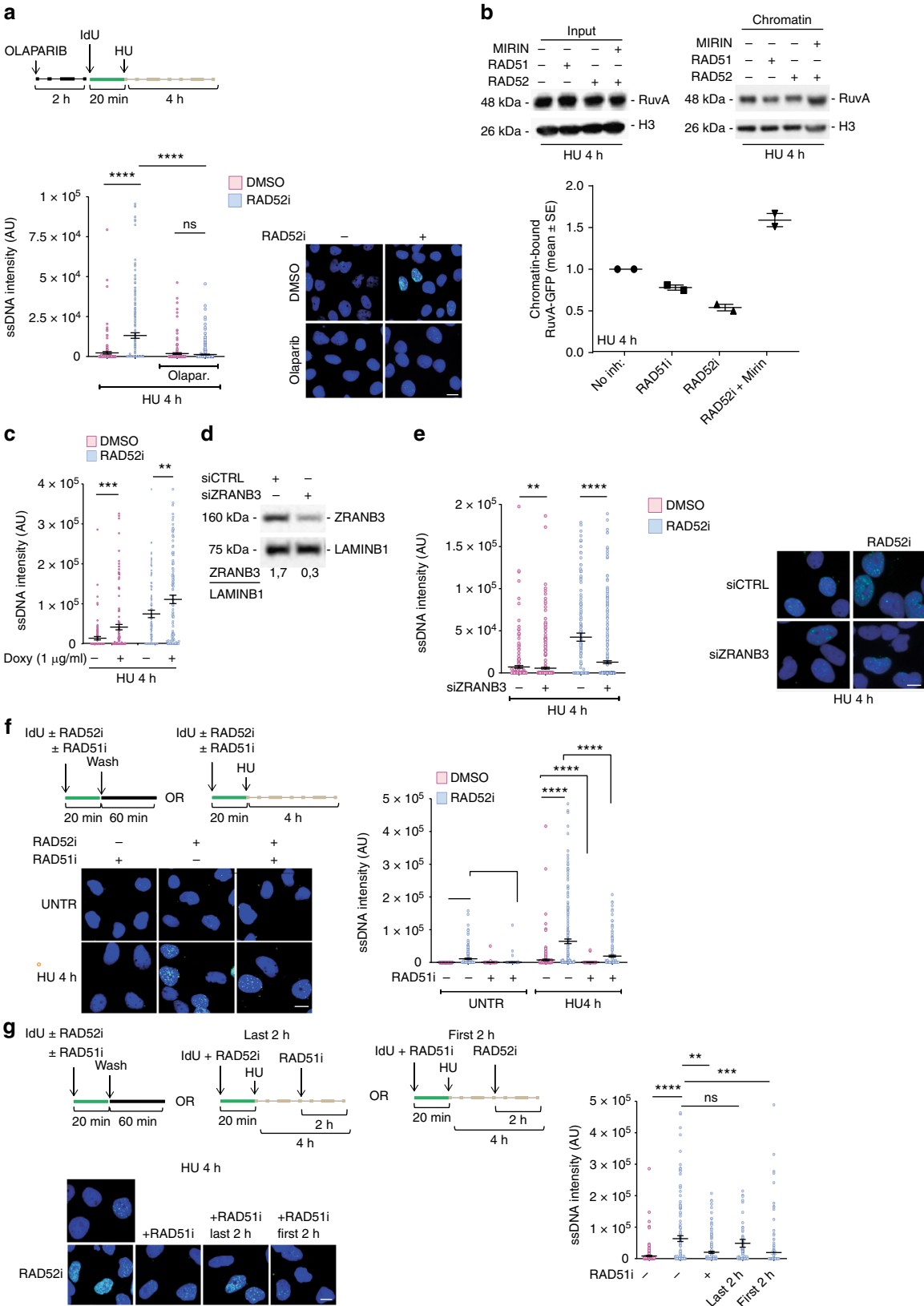

the HJ-like structures of RFs disappear with time in RAD52-inhibited cells. We used ectopically expressed RuvA, which binds HJ and much less Y-shaped DNA structures, as a proxy of the presence of intact RFs[41]. As shown in Fig. 3b, ectopically expressed RuvA-GFP translocated to chromatin after replication arrest. Inhibition of RAD51, which is expected to reduce the number of RFs, decreased the fraction of chromatin-associated RuvA-GFP. Notably, also inhibition of RAD52 decreased the amount of RuvA-GFP in chromatin, which was rescued by MIRIN treatment (Fig. 3b).

To further confirm that nascent ssDNA accumulation occurs downstream RF formation in the absence of RAD52, we

**Fig. 3** Accumulation of nascent ssDNA in RAD52i cells is dependent on fork reversal. **a** Cells were treated as indicated in the scheme. Graph shows the mean intensity of ssDNA staining at least 100 nuclei. Values are presented as means ± SE (ns = not significant; ****$P$ < 0.0001; Mann–Whitney test; $N$ = 475). **b** Analysis of RuvA recruitment in chromatin after replication stress. H3 was used as a loading control. **c** Cells were infected with tetracycline-inducible virus ($V_1$) containing short hairpin RNA (shRNA) sequences direct against SMARCAL1 to produce MRC5 shSMARCAL1 inducible cells lines. Cells were treated or not with doxycicline for 40 h then was labelled with iododeoxyuridine (IdU) to detect nascent ssDNA in presence or not of RAD52 inhibitor and or 2 mM hydroxyurea (HU) for 4 h. Graph shows the mean intensity of ssDNA staining at least 100 nuclei. Values are presented as means ± SE (**$P$ < 0.1; ***$P$ < 0.001; Mann–Whitney test; $N$ = 135). **d** Cells were transfected with scrambled siRNA (siCTRL) or siRNA directed against ZRANB3. Western blot analysis shows level of protein. LAMIN B1 was used as a loading control. **e** Graph shows the main intensity of ssDNA staining for single nuclei from untreated or treated cells. Values are presented as means ± SE (**$P$ < 0.1; ****$P$ < 0.0001; Mann–Whitney test; $N$ = 142). Representative images are given. **f** Analysis of nascent ssDNA in cells inhibited or not for RAD51 and/or RAD52. Graph shows the main intensity of ssDNA staining for single nuclei from untreated or treated cells. Values are presented as means ± SE (****$P$ < 0.0001; Mann–Whitney test; $N$ = 278). **g** Experimental scheme of nascent ssDNA. Analysis of IdU intensity in cells treated or with RAD51 in combination or not with RAD52 inhibitor. When indicated the inhibitors were added at different time point. Dot plots show the mean intensity of ssDNA staining for single nuclei from each cell line after treatment with HU for 4 h. Mean values are represented as horizontal black lines ± SE; $N$ = 164. (ns = not significant; **$P$ < 0.1; ***$P$ < 0.001; ****$P$ < 0.0001; Mann–Whitney test; $N$ = 475). Scale bar represents 5 μm. Source data are provided as a Source Data file

analysed rescue of the RAD52i phenotype in cells depleted of SMARCAL1, ZRANB3 or RAD51, which are crucial for fork reversal[6,8,12,34,42,43]. Surprisingly, depletion of SMARCAL1 by RNAi (Supplementary Fig. 8a) did not affect the formation of nascent ssDNA after replication fork arrest in RAD52-inhibited cells (Fig. 3c). However, concomitant depletion of SMARCAL1 and inhibition of RAD52 resulted in abundant DSBs (Supplementary Fig. 8a, b), suggesting that in SMARCAL1/RAD52-deficient cells, accumulation of nascent ssDNA is driven by end-resection rather than by fork degradation. In contrast, depletion of ZRANB3 substantially reverted accumulation of nascent ssDNA in the RAD52-inhibited cells, while it did not affect the presence of ssDNA in wild-type cells (Fig. 3d, e). Nascent ssDNA accumulation in RAD52-inhibited cells was rescued by reintroduction of an RNAi-resistant ZRANB3 coding sequence (Supplementary Fig. 8c, d). In contrast to SMARCAL1 depletion, knock-down of ZRNAB3 did not stimulate formation of DSBs in RAD52-inhibited cells (Supplementary Fig. 8e, f). This suggests that specific ability of ZRANB3 depletion to affect nascent ssDNA formation in the absence of RAD52 is indeed related to generation and resection of DNA breaks upon SMARCAL1 co-depletion.

RAD51 function is also crucial during fork reversal[7,34]. To confine inactivation of RAD51 to the time of the HU treatment, we used the RAD51 inhibitor B02[33]. While inhibition of RAD52 resulted in a time-dependent formation of nascent ssDNA in response to HU, inhibition of RAD51 prior to fork arrest by HU barely affected the level of nascent ssDNA (Fig. 3f). Interestingly, co-treatment of cells with both inhibitors significantly reduced the amount of nascent ssDNA detected by the native IdU assay (Fig. 3f), but only if RAD51i was added prior to RAD52i and HU, but not if cells were first exposed to RAD52i and HU, and then to RAD51i (Fig. 3g), suggesting that blocking RAD51 function first is crucial to suppress the RAD52i effect.

Collectively, our data indicate that that MRE11-dependent nascent strand degradation in RAD52-inhibited cells occurs downstream of ZRANB3, RAD51 and, possibly, SMARCAL1, all factors involved in fork reversal.

**RAD52 promotes formation of a closed fork structure in vitro.**
Cellular studies reported above indicate that RAD52 interacts with stalled replication forks and prevents their reversal. To determine the mechanism of the RAD52–fork interaction, we employed single-molecule total internal reflection fluorescence microscopy (smTIRFM). Previously, using smTIRFM and a gap DNA decorated with Cy3, a donor of Förster resonance energy transfer (FRET) and Cy5, a FRET acceptor, we showed that

RAD52 interacts with ssDNA by wrapping it around the RAD52 ring[44]. The RAD52 oligomeric ring has two DNA-binding sites: a primary ssDNA-binding site in the grove spanning the ring circumference[45–47] and a second DNA-binding site that can bind either ssDNA or dsDNA[48]. Since the structure of the stalled replication fork contains both ssDNA and dsDNA, RAD52 may bind both simultaneously. In addition, through an RQK motif RAD52 also interacts with RPA[49], which is also present at stalled replication forks that contain ssDNA. To determine whether RAD52 can configure stalled replication forks so that they become inaccessible to fork remodelers, we analysed its interaction with the gap DNA (G1) and with the three substrates mimicking stalled replication fork (Fig. 4 and Supplementary Fig. 9). First, using electrophoretic mobility shift assays (EMSA), we confirmed that RPA and RAD52 form a stable ternary complex with both the gap (G1) and the fork DNA (RF2) (Fig. 4a, b).

We then applied smTIRFM to monitor the conformations of the replication fork in the presence of RAD52 and RPA, as well as the fork dynamics. The G1 and the RF1 structures were identical except for the presence of the leading strand dsDNA arm on the RF1 (Supplementary Fig. 9). The Cy3 and Cy5 dyes flanking the 30 nt ssDNA region allowed us to monitor in real time the conformational states of the ssDNA region of the two substrates. RF2 (Fig. 4c) had the same structure as RF1, but the Cy5 dye was moved to the end of the leading stand arm. FRET between the Cy3 and Cy5 fluorophores on RF2 reflected the relative positions of the leading and lagging arms. Similarly, RF3 (Fig. 4d) was reporting on the conformational arrangement of the "lagging" arm and the parental duplex. The Cy3/Cy5 labelled DNA substrates were tethered to the surface of the smTIRFM flow cell and incubated with 1 nM RPA and various concentrations of RAD52. Several hundred of individual DNA molecules were observed in each experiment and subjected to two types of analysis. First, FRET values from all surface-tethered molecules were collectively analysed to yield FRET distributions. Second, the forks' conformational dynamics was observed in FRET trajectories (time-based changes in FRET originating from each substrate-containing spot on the flow cell surface). Under conditions selected for our experiments, the ssDNA is compacted yielding high FRET for G1 and RF1 substrates. RPA extends the ssDNA in the gap resulting in a low FRET. FRET distributions for both, the protein-free and RPA-bound G1 and RF1 contain single Gaussian peaks. No changes in FRET were observed in the individual trajectories of G1 or RF1 alone, and very infrequent changes in FRET were observed in the presence of RPA. Addition of RAD52 shifted the FRET distribution and induced dynamic conformational changes in G1 and to a lesser extent in RF1,

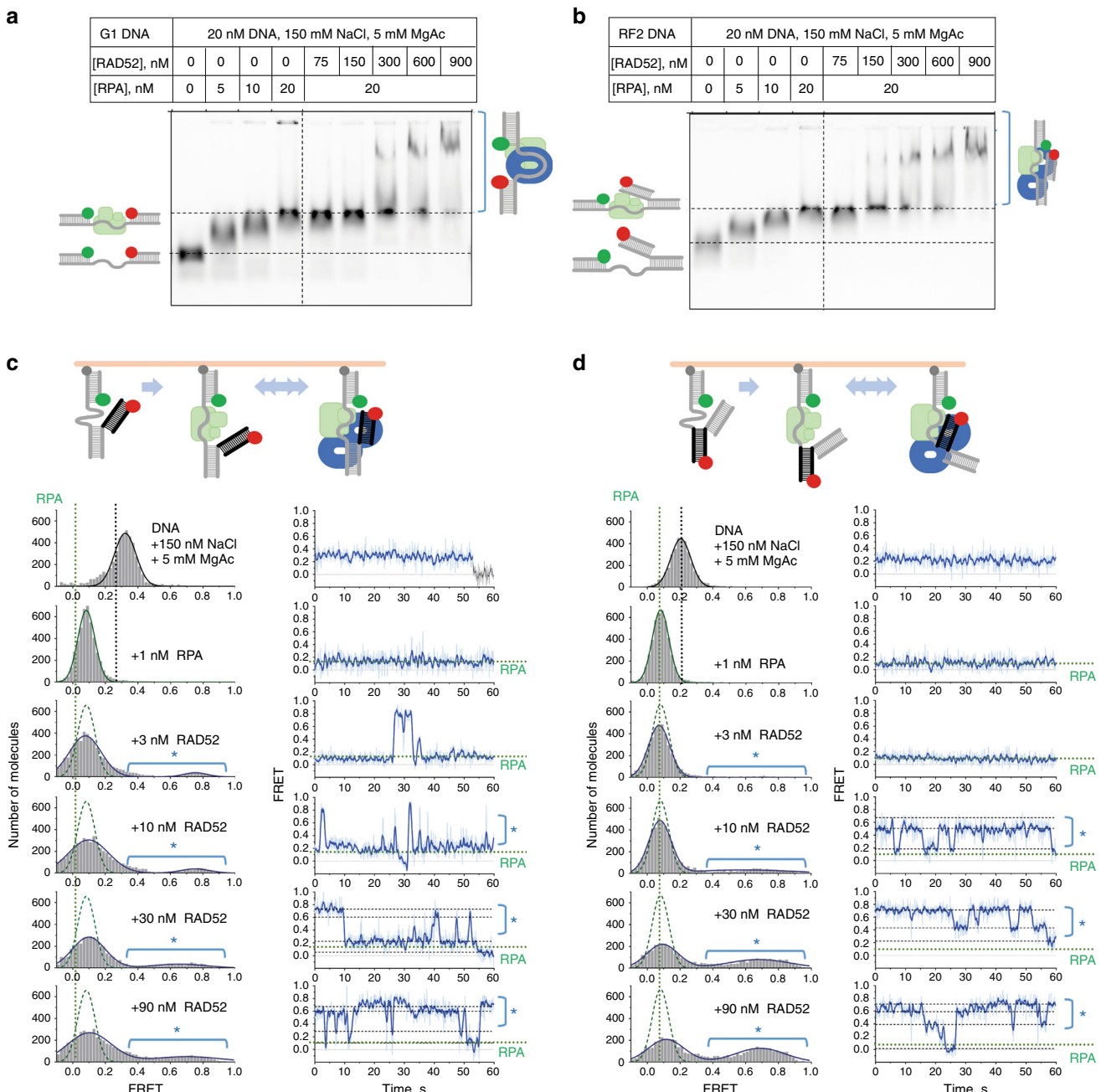

**Fig. 4** RAD52 promotes formation of a reversal-resistant fork structure. **a**, **b** Electrophoretic mobility shift analysis of the RPA and RAD52 binding to G1 DNA (**a**) and a model replication fork, RF1 (**b**). DNA substrates were incubated with the indicated concentrations of RPA and RAD52. The complexes and the unbound DNA were then separated by electrophoresis in 0.8% TAE agarose. **c**, **d** Single-molecule analysis of the replication fork-like structures in the presence of RPA and RAD52. RF2 (**c**), which reports on the conformation and motion of the leading strand (black) arm relative to the lagging strand arm tethered to the surface, and RF3 (**d**), which reports on the relative position and dynamics of the parental duplex (black) relative to the surface-tethered lagging strand arm, were immobilised on the surface of the TIRFM flow cell. The Cy3 dye (green circle) was excited by 532 nm TIR illumination, while the Cy5 dye (red circle) was excited via Förster resonance energy transfer (FRET) from Cy3. Fluorescence of the Cy3 and Cy5 dyes was recorded separately and used to calculate FRET efficiency. FRET distributions in the left column of each panel were obtained from combining three short movies and represent the overall FRET states of each substrates under the indicated conditions. Dotted green lines show the FRET distribution for each substrate in the presence of RPA only overlaid over the distribution in the presence of RPA and RAD52. Regions of the distributions marked by a blue star indicate states where the Cy5-labelled arm is brought close to the Cy3-labelled lagging strand arm. Movies recorded for 1 min were used to evaluate the conformational dynamics of the RF2 and RF3 under each experimental condition. Representative single-molecule FRET trajectories are shown in the right column by their respective FRET distributions. Source data are provided as a Source Data file

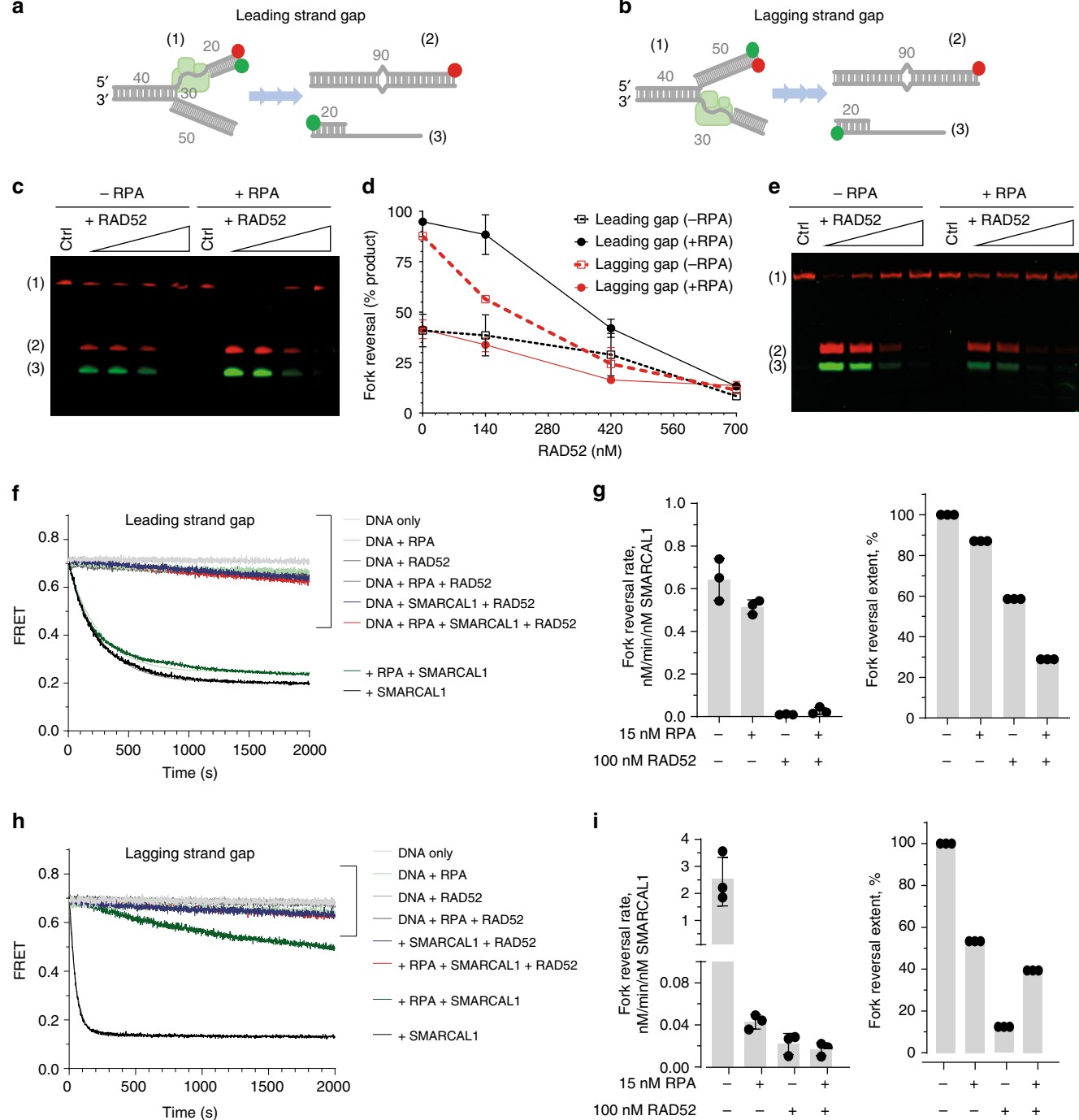

**Fig. 5** RAD52 prevents fork reversal by SMARCAL1. **a, b** Cartoon depiction of the DNA substrates and the reversal reaction employing synthetic fork with a 30 nt gap on the leading (**a**) and lagging (**b**) strand, respectively. The lengths of the dsDNA and ssDNA features are marked in grey. Green circles depict Cy3 dyes, red circles depict Cy5 dyes. The substrate and the products of the fork reversal reactions are separated on the agarose gels after deproteinization. All reactions were carried out in the presence of 3 nM of forked DNA. **c–e** Representative gels and quantification of the fork reversal reactions by 20 nM SMARCAL1 in the presence of increasing concentrations of RAD52. Reactions were initiated by addition of SMARCA1 and stopped at 15 min. Only Cy5 channel data were used for quantification. **f** Representative fork reversal time courses for the DNA substrate containing a leading strand gap. Solid faint lines behind the experimental data indicate fits to a single exponential. **g** The rates of the fork reversal reactions and their extents were calculated by fitting the data to exponential decay functions as described in the Methods section. The resulting rates are shown for three independent experiments. **h, i** The same as **f, g**, but for the substrate with a lagging strand gap. Source data are provided as a Source Data file

suggesting that RAD52 interacts with the RPA-bound ssDNA within these two substrates. RAD52 addition to the RPA-bound RF2 and RF3 resulted in a high FRET peak, suggesting that both the leading strand arm and the parental duplex are brought close to the lagging strand arm. Individual trajectories showed a highly

dynamic behaviour with each fork transitioning between different FRET states, which were higher than those observed for the RPA-bound fork, suggesting a more compact conformation. Notably, the RAD52 binding to the G1, FR1, RF2 and RF3 substrates did not completely displace the RPA bound to the ssDNA region.

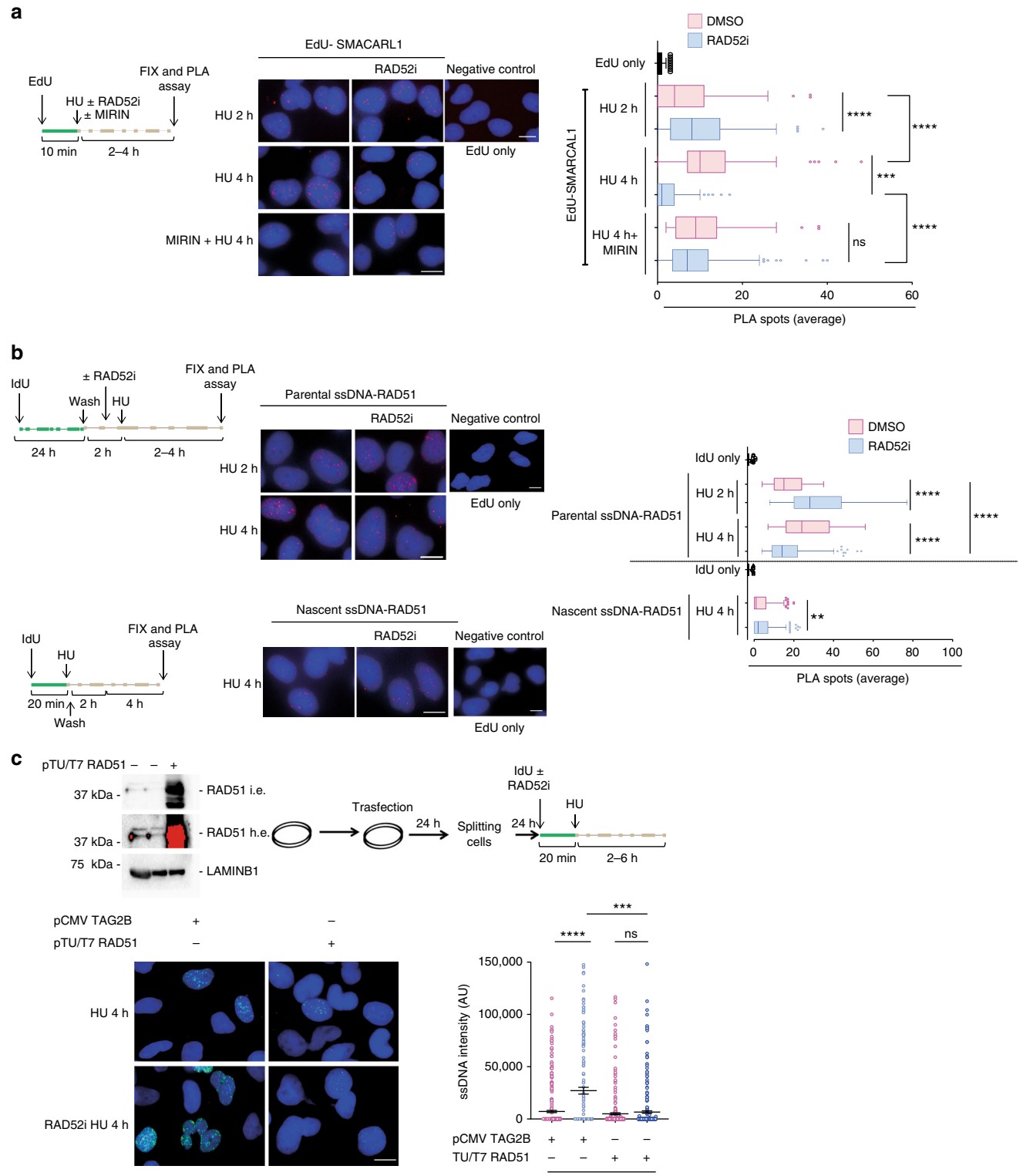

EMSAs (Fig. 4a, b) confirmed the presence of both RAD52 and RPA on these substrates.

Collectively, these results indicate that RAD52 can bind to the model fork substrates, remodelling the fork into a structure potentially refractory to reversal.

**RAD52 antagonises fork reversal by SMARCAL1.** SMARCAL1 is a DNA-dependent ATPase, which displays an annealing activity[50] and remodels three- and four-way DNA junctions by

promoting branch migration and fork reversal[42]. While SMARCAL1 interacts with and reverses replication fork-like structures with ssDNA gaps on either leading or lagging strand, interaction with RPA potentiates reversal of forks with the leading strand gap and inhibits reversal of lagging strand gap containing forks[51]. Here, we carried out in vitro fork reversal experiments and probed the effect of RAD52 on the SMARCAL1-mediated fork reversal (Fig. 5 and Supplementary Fig. 10). The design of our DNA substrates was based on that by Betous et al.[51] (Fig. 5a, b).

**Fig. 6** RAD52 affects fork recruitment of SMARCAL1 and RAD51. **a** Analysis of SMARCAL1 fork recruitment by in situ EdU-proximity ligation assay (PLA). Cells were treated as indicated. The graph shows the number of PLA spots per cell. As a negative control for the PLA, cells were incubated with only the anti-biotin antibody (EdU only). Values are presented as means ± SE (ns not significant; ***$P < 0.001$; ****$P < 0.0001$; Mann–Whitney test; $N = 145$). Representative images are shown. **b** Analysis of nascent and parental iododeoxyuridine (IdU)-RAD51 PLA. Cells were treated as in the scheme. The graph shows the number of PLA spots per cell ± SE. As a negative control for the PLA, cells were incubated with only the anti-IdU antibody (IdU only). Values are presented as means ± SE (**$P < 0.01$; ****$P < 0.0001$; Mann–Whitney test. Nascent DNA $N = 170$; Parental ssDNA $N = 379$). Representative images are shown. **c** Analysis of nascent ssDNA formation in RAD51 overexpressing cells. On top, western blot analysis showing RAD51 overexpression and the experimental scheme used. The graph shows the intensity of ssDNA staining in at least 100 nuclei from two independent experiments. Values are presented as means ± SE (ns not significant; ***$P < 0.001$; ****$P < 0.0001$; Mann–Whitney test; $N = 436$). Representative images are shown. Scale bar represents 5 μm. Source data are provided as a Source Data file

The effect of RPA on the SMARCAL1 activity was the same as previously reported[51,52]. Namely, positioning of the RPA DNA-binding domains A and B close to the fork junction, as occurs on the leading strand gap, facilitated fork remodelling by SMARCAL1[52]. In the absence of RPA, the extent of the reversal reaction reached 100 and 45% in the presence of the stoichiometric amounts of SMARCAL1 for the lagging strand and leading strand gap substrates, respectively. RPA stimulated reversal of the leading strand gap containing substrate and inhibited that of the substrate with the lagging strand gap (Supplementary Fig. 10d–f)[51,52]. Addition of RAD52 antagonised reversal of both substrates in the presence and absence of RPA (Fig. 5c–e). The same trend was observed in the FRET-based fork reversal experiments, which followed time-based conversion of the substrate (high FRET) into the products (low FRET) (Fig. 5f–i). RAD52 affected both the rate and the extent of the fork reversal reaction (Fig. 5g–i) consistent with the model where RAD52 remodels the RPA-containing fork into a structure refractory to reversal by SMARCAL1.

**RAD52 inhibition affects loading of fork reversal factors.** Having demonstrated that loss of RAD52 leads to nascent strand degradation downstream fork reversal and that, in vitro, RAD52 prevents fork reversal by SMARCAL1, we investigated whether inactivation of RAD52 could stimulate fork recruitment of SMARCAL1. To this end, we performed EdU-PLA at different times of HU treatment. As expected, SMARCAL1 was readily found associated with nascent DNA after replication arrest and this association increased over time (Fig. 6a). Interestingly, inhibition of RAD52 significantly increased the presence of SMARCAL1 at stalled forks at 2 h of HU (Fig. 6a), as well as the presence of SMARCAL1 nuclear foci (Supplementary Fig. 11a). Surprisingly, RAD52 inhibition led to a decrease of SMARCAL1 at perturbed forks at 4 h of HU, which is dependent on the excessive nascent strand degradation as it was reverted by MIRIN (Fig. 6a). Consistent with this observation, MIRIN rescued also the presence of SMARCAL1 in chromatin after RAD52 inhibition (Supplementary Fig. 11b). Moreover, in RAD52-inhibited cells, also more RAD51 nuclear foci were detected (Supplementary Fig. 11c) and increased recruitment of ZRANB3 at nascent DNA was visualised by EdU-PLA upon replication fork arrest (Supplementary Fig. 12a, b).

Next, we investigated whether inhibition of RAD52 could affect recruitment of RAD51 to fork. As RAD51 may be found at parental or nascent ssDNA to prevent fork degradation, we performed native IdU-PLA experiments after labelling of either the parental or the nascent strand. Inhibition of RAD52 increased the presence of RAD51 at the parental ssDNA at 2 h of HU while decreased its amount at 4 h, while the amount of RAD51 interacting with the nascent ssDNA was barely affected (Fig. 6b). Hence, we asked whether excessive fork degradation, observed in the absence of RAD52, was due to exhaustion of the RAD51 pool by too much fork reversal. We overexpressed RAD51 in cells

treated or not with the RAD52i and HU before analysing the presence of nascent ssDNA (Fig. 6c). Interestingly, while overexpression of RAD51 in mock-inhibited cells did not change the amount of ssDNA exposed at nascent strand of blocked replication forks, it clearly reduced the formation of nascent ssDNA in the presence of the RAD52 inhibitor (Fig. 6c).

Altogether, these results indicate that RAD52 plays a role in controlling recruitment of fork reversal enzymes to perturbed replication forks and that degradation of nascent strand occurs because of the inability of endogenous pool of RAD51 to protect all RFs.

**RAD52 prevents persistence of unreplicated DNA after HU.** Next, using a DNA fibre assay, we investigated whether MRE11-dependent degradation of nascent strand could affect the ability of RAD52-inhibited cells to recover from replication arrest (Fig. 7a). As expected after a short HU treatment, the majority (78%) of stalled replication forks restarted in wild-type cells, however, RAD52 inhibition resulted in a mild reduction of restarting forks (Fig. 7a). Inhibition of MRE11 in RAD52-inhibited cells caused a substantial reduction in the ability to resume replication (Fig. 7a). Of note, the mild defect in replication fork recovery conferred by RAD52 inhibition was further affected by treatment with the CDC7i XL413, which blocks firing of new origins (Fig. 7a). These results suggest that, in the absence of RAD52, a fraction of restarting replication forks are indeed dormant origins activated to rescue stalled forks. Thus, we analysed whether unreplicated regions might persist during recovery from replication fork arrest in cells treated with RAD52i. To this end, we used the presence of parental ssDNA as readout of a partially replicated template. As expected, at 24 h of recovery, little parental ssDNA was detectable in control cells (Fig. 7b). In contrast, parental ssDNA accumulated during recovery in cells that were treated with HU and the RAD52i (Fig. 7b). Interestingly, in RAD52i-treated cells, the amount of exposed parental ssDNA decreased if RAD51 was inhibited before fork stalling, but greatly increased if RAD51 was inhibited during recovery (Fig. 7b).

Consistent with this result, inhibition of RAD52 during replication fork arrest resulted in a subsequent significant increase of the number of RAD51-positive cells during the recovery (Fig. 7c).

Collectively, these results indicate that RAD52 is essential for a proper recovery of stalled replication forks and that, in its absence, a fraction of the under-replicated DNA that persist is channelled to a RAD51-dependent mechanism to complete duplication.

Having demonstrated that a RAD51-dependent pathway is engaged to complete replication at regions of fork degradation in the RAD52-inhibited cells, we sought to determine if persistence of unreplicated regions may continue into mitosis. Thus, we analysed the presence of ultra-fine DNA bridges (UFBs) by anti-Bloom Syndrome helicase (BLM) immunofluorescence in

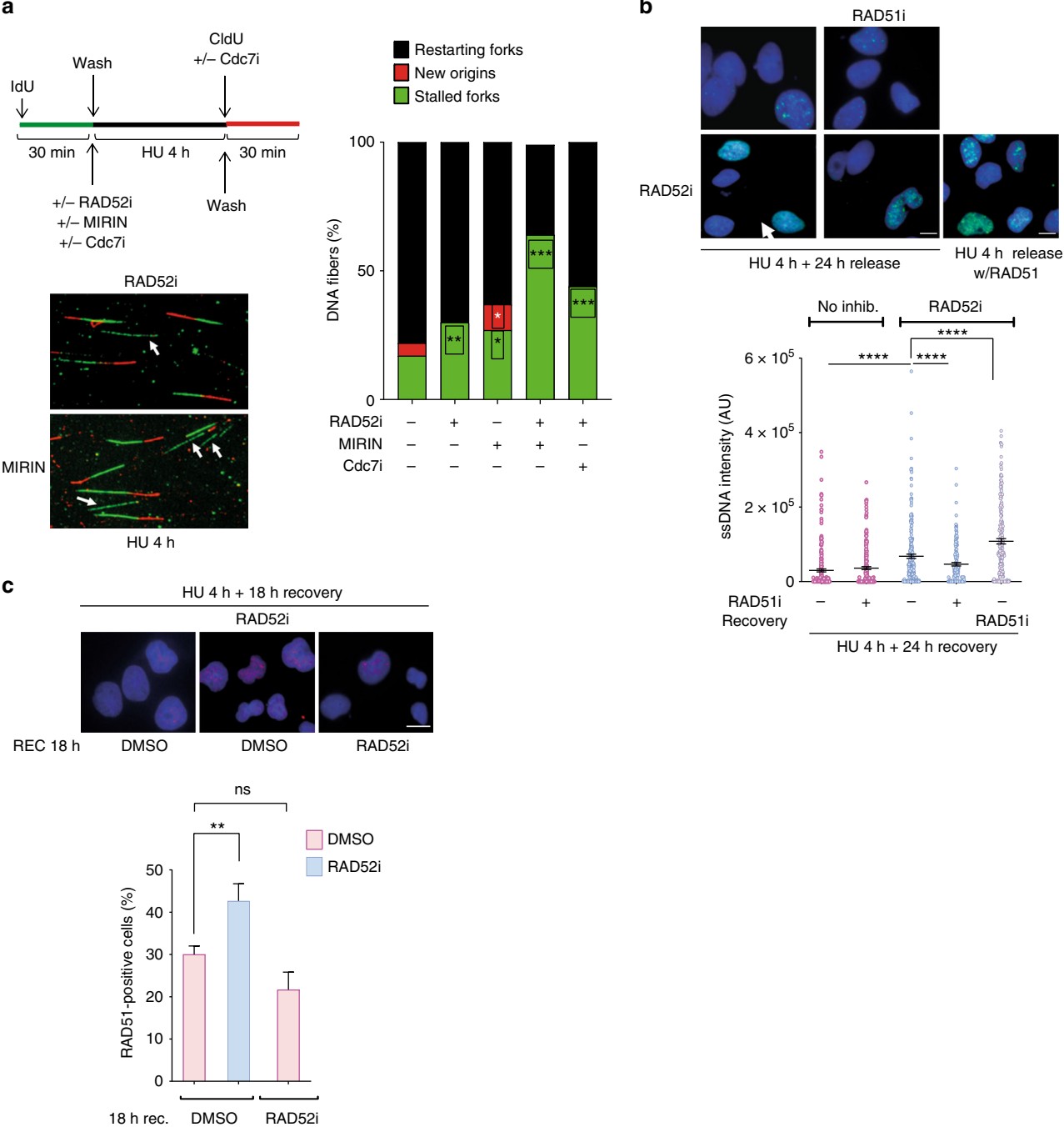

**Fig. 7** Loss of RAD52 leads to persistence of unreplicated DNA. **a** Analysis of replication fork restart using DNA fibre assay. Cells were treated as indicated in the scheme on the top. The graph shown the percentage of each event. Representative images of DNA fibres fields are presented. Arrows denote stalled forks. (*$P < 0.05$; **$P < 0.01$; ***$P < 0.001$; ANOVA test; $N = 140$). **b** Analysis of the persistence of parental ssDNA (template gaps) during recovery from replication arrest. Cells were treated as indicated and analysed by the native iododeoxyuridine (IdU) assay. The graph shows the intensity of ssDNA staining in at least 100 nuclei from two independent experiments. Values are presented as means ± SE (****$P < 0.0001$; Mann–Whitney test; $N = 141$). Representative images are shown. **c** Analysis of RAD51 nuclear foci. Cells were treated with 2 mM hydroxyurea (HU), in the presence or not of the RAD52i, for 4 h and recovered as indicated, without or with RAD52i. The graph shows the percentage of nuclei with RAD51 foci. Data are presented as mean ± SE from three independent experiments. (ns, not significant; **$P < 0.01$; ANOVA test; $N = 145$). Representative images are shown. Scale bar represents 5 μm. Source data are provided as a Source Data file

anaphase cells after recovery from replication arrest (Fig. 8a). As expected, a low level of UFB-positive anaphases was detected in control cells after recovery (Fig. 8b). In contrast, cells experiencing replication fork arrest in the absence of RAD52 showed a substantial increase in the number of anaphase cells with UFBs, as well as in the number of UFBs per anaphase, which were

often two (Fig. 8b). Notably, about half of the UFBs detected in anaphases from RAD52-inhibited cells stained positively for the RPA32, indicating that they still contain under-replicated, ssDNA, regions (Fig. 8c). The presence of UFBs was reduced if RAD51 was inhibited before fork arrest in RAD52i-treated cells (Fig. 8c). RAD51 inhibition during recovery resulted in a strong

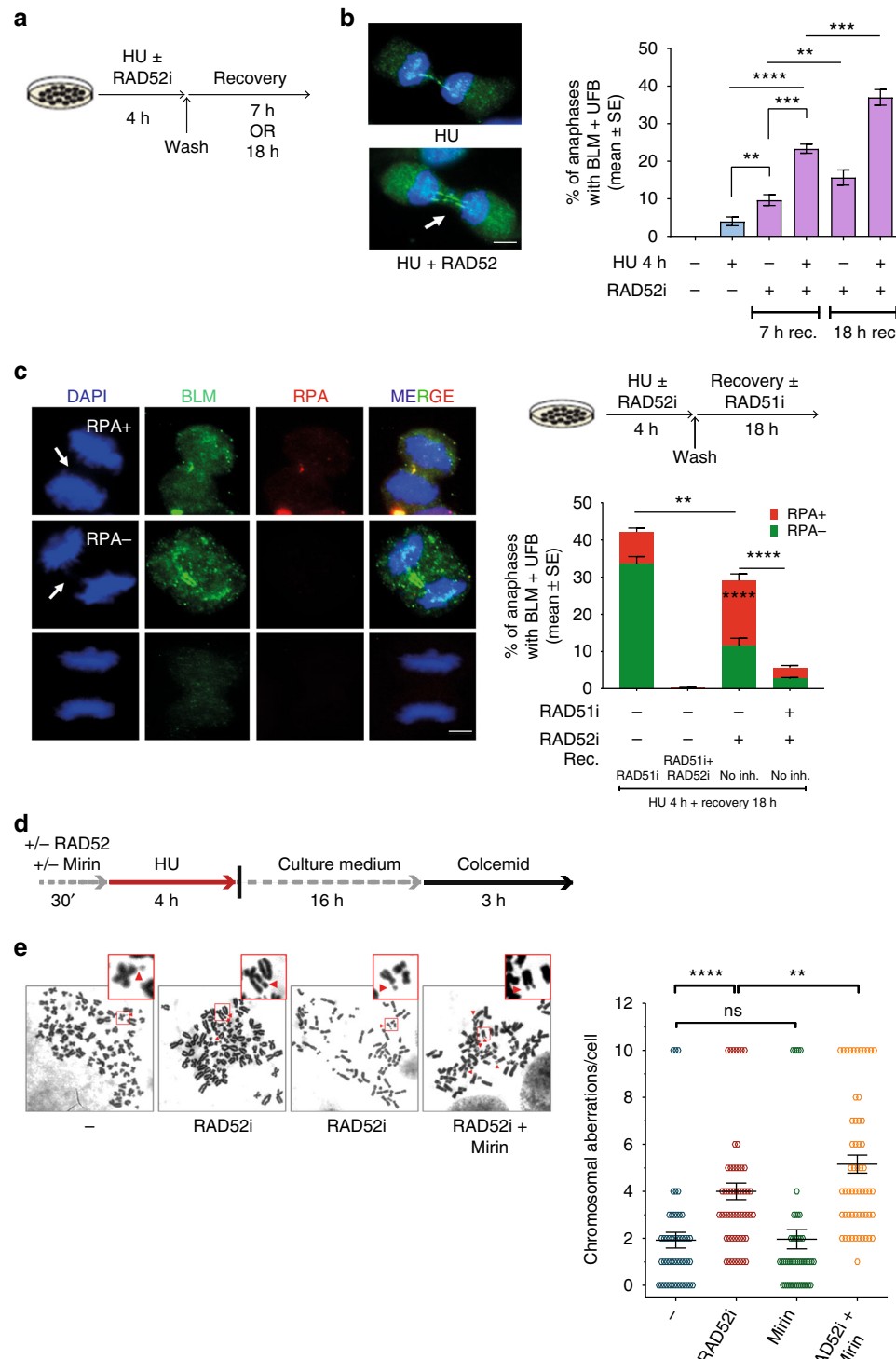

**Fig. 8** Loss of RAD52 leads to enhanced genome instability. **a** Experimental scheme used for analysis of UFBs. **b** Quantification of BLM-positive UFBs in cells treated as in **a**. The graph shows the percentage of anaphases with BLM-positive UFBs. Representative images of anaphase cells with BLM-positive UFB (green) are shown on the right. (**P<0.1; ***P < 0.001; ****P < 0.0001; Mann–Whitney test; N = 50). **c** Analysis of UFBs in cells treated as shown in the experimental scheme on the top. Quantification of anaphase/telophase cells with BLM/RPA-positive UFBs. The graph shows the percentage of anaphases with BLM-positive UFBs. Representative images of an anaphase cell stained with BLM (green) as control, RPA (red), and DAPI (blue) are shown in the bottom of the panel. Minimum of 50 ana/telophase cells were scored per experiment; arrows indicate position of UFB. (**P < 0.01; ****P < 0.0001; Mann–Whitney test). **d**, **e** Analysis of chromosomal aberrations in cells treated as shown in the experimental scheme. Dot blot shows the number of chromosomal aberrations per cell. Data are presented as means of three independent experiments. Horizontal black lines represent the mean ± SE (ns not significant; **P < 0.01; ****P < 0.0001; two-tailed Student's t-test; N = 57). Representative images of Giemsa-stained metaphases are given. Arrows indicate chromosomal aberrations. Insets show an enlarged portion of the metaphases for a better evaluation of chromosomal aberrations. Scale bar represents 5 µm. Source data are provided as a Source Data file

accumulation of UFB-positive anaphases in wild-type cells, mostly RPA negative (Fig. 8c).

We next analysed if the persistence of under-replicated DNA could correlate with chromosome instability (Fig. 8d). The absence of RAD52 during replication fork arrest stimulated a significant increase in chromosomal aberrations (Fig. 8e). Inhibition of RAD51 during recovery of RAD52-inhibited cells aggravated the phenotype with accumulation of many aberrant mitosis that resemble mitotic catastrophe making unreliable the analysis of chromosome damage (Supplementary Fig. 13). Concomitant inhibition of MRE11 and RAD52 during replication fork arrest resulted in more chromosome damage, compared with the RAD52-inhibited cells (Fig. 8e).

Collectively, these results indicate that RAD52 is required for correct restart after fork arrest and that, in its absence, a RAD51-dependent mechanism is activated to attempt the replication of under-replicated regions. Moreover, our results suggest that the regions left under-replicated because of RAD52 inhibition may persist up to mitosis triggering genome instability.

## Discussion

Here, we show that RAD52 contributes to the stability of nascent strand after replication fork arrest by binding to the replication fork and preventing an excessive engagement of fork reversal enzymes.

Such previously unknown fork-protective role of RAD52 is unrelated to either MUS81-dependent DSB formation, BIR or MIDAS[11,23,24,27,35]. Instead, we show that RAD52 can associate with the replication fork early after HU treatment and, most importantly, that it can be found at both parental and nascent ssDNA. We observe that RAD52 association with nascent ssDNA after fork arrest can be modulated interfering with formation of RFs, thus, RAD52 could bind to the ssDNA formed at paired nascent strand of RFs. However, RAD52 inhibition does not affect much the presence of RAD51 at the nascent ssDNA after fork arrest as one would expect if RAD51 nucleofilaments were destabilised or not assembled properly. This is in line with BRCA2 being the main human recombination mediator[20–22,31]. So, it is unlikely that nascent ssDNA-bound RAD52 plays a relevant role in stabilising RAD51 nucleofilaments once they have been assembled by BRCA2.

MRE11-dependent degradation of nascent strand after replication perturbation is a pathological event repeatedly observed in BRCA2-deficient cells and in other mutants of fork-protecting factors[6,7,10,11,14,15,30,38]. Exonucleolytic degradation of nascent strand in the absence of BRCA2 is unidirectional[11]. Indeed, the unaffected nascent strand is cleaved by the MUS81 complex as a long 3′ flap and repetitive cycles of fork reversal/degradation/ cleavage have been proposed to account for the extensive loss of nascent DNA observed with fibre assay[11]. As RAD52 and MUS81 cooperate to resolve demised replication forks[23,27,35], it is very likely that inhibition of RAD52 results in a complex phenotype whereby the 5′-ended nascent strand is degraded but the complementary strand (i.e., the 3′-flap) cannot be cleaved by MUS81. This dual function of RAD52 might account for the presence of apparently intact replication tracks, which contain extensive regions of ssDNA, as indicated by the S1-fibre assay. Although detection of ssDNA by native IF in the nascent has been correlated to RF degradation[6], we cannot correlate amount of native IdU fluorescence with length of degraded DNA, since each ssDNA focus likely derives from multiple replication forks undergoing processing in each replication factory.

Loss of BRCA2 leads to fork destabilisation as RAD51 cannot properly loads onto the nascent ssDNA formed at the RF either spontaneously, or after a controlled degradation by DNA2[53].

Although RAD52 is found at nascent ssDNA, its absence does not affect the interaction of RAD51 with nascent DNA. Most likely, RAD52 is found at the nascent ssDNA because of the reported function as a loader of MRE11 at the RF[7]. From this point of view, the presence of a robust MRE11-dependent activation, even when RAD52 is inhibited, implies that cells possess multiple ways to activate this fork restart mechanism, followed by several means to cleave the demised fork even in the absence of the MUS81 complex[41].

In contrast, association of RAD52 with the parental ssDNA likely occurs independently of fork reversal. Indeed, our single-molecule experiments indicate that RAD52 binds to a model fork and promotes closure of the Y-shaped structure around the protein bringing all three arms of the fork together, which may prevent further fork remodelling and contribute to the fork stabilisation while cells wait for a safer restart. Previous studies showed that excessive fork reversal performed by SMARCAL1, as well as the loss of the RAD51 antagonist RADX also causes DNA damage[15,16,54,55]. Thus, RAD52 may serve as a gatekeeper to limit reversal of stalled forks or at least of a subset of them. Indeed, inhibition of RAD52–ssDNA interaction increases the amount of SMARCAL1 associated with blocked fork as well as relocalisation of RAD51[7,34].

Accordingly, in RAD52i cells, accumulation of nascent ssDNA likely takes place at RFs as consistently demonstrated for BRCA2-deficient cells[6,7]. Indeed, inhibition of RAD51 before that of RAD52, pre-treatment with Olaparib or depletion of ZRANB3 effectively reduces nascent strand degradation in RAD52-inhibited cells. Such unscheduled and excessive fork reversal exhausts the endogenous pool of RAD51 that is available for fork protection, and thus overexpression of RAD51 in RAD52-inhibited cells is sufficient to revert their fork degradation phenotype. Many proteins participate in fork protection by stabilising RAD51 at the RF[3,4]. Of note, both RAD52 and SMARCAL1 recognise the same RPA-binding site[5,49,56,57], suggesting a potential competitive mechanism of recruitment. Our results are the first demonstration of a mechanism that remodels the fork structure to limit reversal and degradation.

Inhibition of RAD52 leads to a net reduction of SMARCAL1 at nascent DNA at a late time point, which is apparently at odds with the increase observed early after HU. Since MIRIN can revert this effect, our results indicate that SMARCAL1 is properly loaded in absence of RAD52 to execute fork reversal but is then dislodged because of unscheduled fork degradation. Such run-off of SMARCAL1 may underestimate its fork recruitment after RAD52 inhibition and, most importantly, could relate with the less efficient fork recovery of replication as SMARCAL1 is also important to resume stalled forks[12,56].

In the absence of RAD52, the number of restarting forks is reduced by MIRIN treatment, suggesting that the type of intermediate generated by MRE11 supports restart, which may involve repriming events similarly to what occurs in bacteria with PriA acting downstream RecBCD[58,59].

However, parental ssDNA can be observed in cells exposed to HU and the RAD52 inhibitor even at long recovery times suggesting that a portion of the genome remains under-replicated. The accumulation of under-replicated DNA has been also reported in the presence of DNA damage and can engage RAD51 in post-replication repair[8]. Consistent with this, cells recovering from replication arrest and RAD52 inhibition engage much more RAD51 in nuclear foci. Most importantly, inhibition of RAD51 during recovery increases persistence of parental ssDNA if stalled forks have been processed in the presence of the RAD52 inhibitor.

Consistent with this observation, inhibition of RAD51 during recovery in cells that experienced fork stalling in absence of active

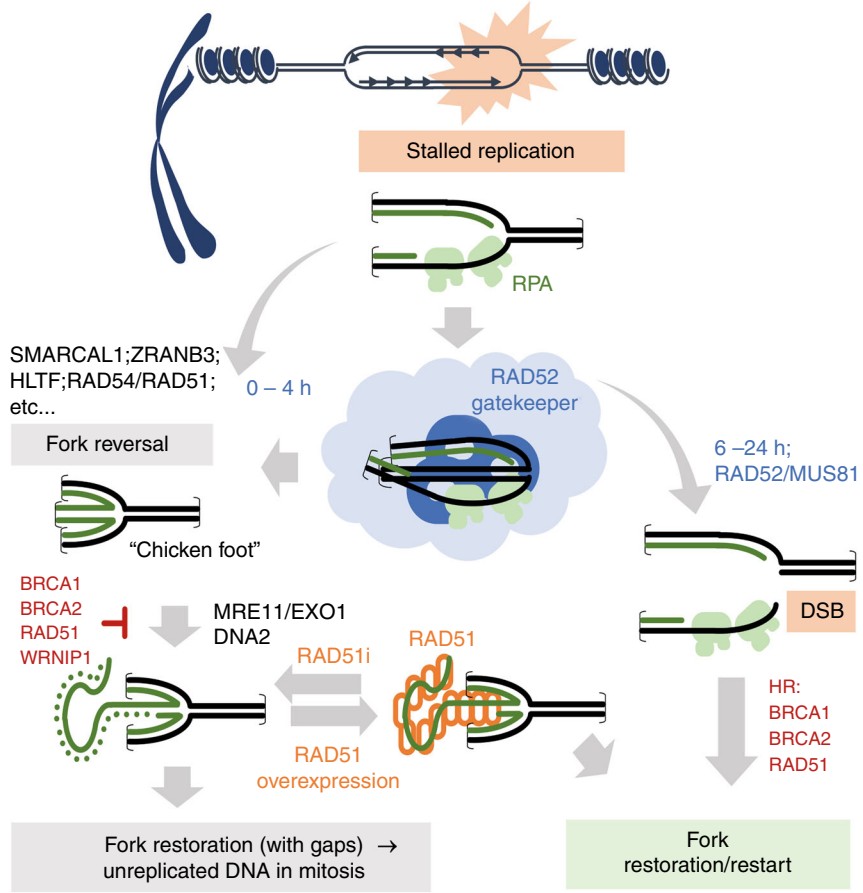

**Fig. 9** Multiple roles of RAD52 at perturbed replication forks. Model representing multiple roles of RAD52 at perturbed replication forks. In response to replication fork perturbation, RAD52 plays crucial roles both upstream replication fork remodelling and downstream, when integrity of remodelled forks is undermined. The role upstream replication fork remodelling is crucial to limit reversal of replication fork, which may be deleterious if unscheduled. These two independent roles of RAD52 are carried out also through distinct protein partners, and may be relevant under either normal or pathological replication perturbation (see text for details)

RAD52 greatly affects viability. RAD51-dependent post-replication repair may involve BRCA2 as a mediator[8]. Therefore, synthetic lethality between BRCA2 and RAD52 might derive not only from loss of activation of MUS81, as proposed[27], but also from the essential role of RAD51 in supporting viability of RAD52-deficient cells.

In conclusion, we propose a novel model describing the function of RAD52 at perturbed replication forks (Fig. 9). After replication fork stalling, RAD52 is recruited at perturbed forks to limit accessibility of fork remodelling factors, avoiding excessive fork reversal and inability of RAD51 to subsequently stabilise them. Later, RAD52 may also contribute to stabilisation of RAD51 filaments at the reversed replication forks.

In the absence of RAD52, or when its association with ssDNA is inhibited, more fork reversal enzymes gain access to stalled replication forks. Accumulation of RFs leads to deprotection of a subset of them and MRE11-dependent degradation. In the absence of RAD52, the remaining nascent ssDNA strand cannot be cleaved by MUS81. While, this intermediate may be reprimed leading to fork recovery, this restoration may be inefficient resulting in persistence of under-replicated DNA and RAD51-dependent post-replication repair/template switching, which ensure viability and reduce the chromosome instability otherwise associated with inhibition of RAD52. It remains to be clarified why depletion or knock-out of RAD52 has never provided significant DNA damage-related phenotypes[25]. Our data confirm

that loss of RAD52 does induce a moderate chromosome instability and suggest that most of the "rad52" phenotype may be masked by a concomitant increase in the function of RAD51.

The RAD52 gatekeeper role at the perturbed replication forks, which we describe here, may be also relevant to several types of cancer. For example, overexpression of RAD52 may prevent excessive fork reversal making cancer cells differentially sensitive to some anticancer drugs. On the other hand, the gatekeeper function of RAD52 at replication forks may be critical in the absence of BRCA1 or BRCA2, which may further explain synthetic lethality between BRCA defects and RAD52 depletion or inhibition. Moreover, understanding the RAD52 function in the context of fork reversal may provide strong mechanistic basis for the validation of RAD52 inhibitors in therapy.

## Methods

**Cell lines and culture conditions**. The MRC5SV40 cells were maintained in Dulbecco's modified Eagle's medium (DMEM; Life Technologies) supplemented with 10% foetal bovine serum (Boehringer Mannheim) and incubated at 37 °C in a humidified 5% $CO_2$ atmosphere. To generate stable shRAD52, MRC5SV40 were transduced with lentivirus expressing two different shRNA sequences (Sigma-Aldrich Mission lentivirus, sequence codes 271352 ($V_2$) and 271415 ($V_1$)). To generate the inducible shSMARCAL1 cells, MRC5SV40 were transduced with lentivirus expressing an shSMARCAL1 cassette under the control of a Dox-regulated promoter at 0.5 of multiplicity of infection (MOI) (Dharmacon Smart-Vector inducible lentivirus, sequence code V3SH11252-227970177). After puromycin selection at 300 ng/ml, a single clone was selected and used throughout

the study. Cell lines were routinely tested for mycoplasma contamination and maintained in cultures for no more than one month.

**Oligos and plasmids**. pCMV-tag2B (pFLAG) empty vector was used to produce pCMV-tag2B-RAD52 (pFLAG-RAD52) in our laboratory. RAD52 was amplified using the following primers and restriction enzymes:

5′-TTAGAATTCAGAAGGAGATATACCATGGGCAGC-3′ primer Fwd RAD52 (EcoR1)

5′-TAACTCGAGCTTTGTTAGCAGCCGGATCC-3′ primer Rev RAD52 (XhoI).

MUS81 expression was knocked-down by transfection with specific siRNA: Hs MUS81 6 FlexiTube siRNA cat # SI04300877. ZRANB3 was downregulated by transfection with ZRANB3 siRNA: siGENOME siRNA D-010025-03-005 Dharmacon cat # 84083. The pMSC-FLAG-ZRANB3 and ZRANB3 siRNA were a gift from Prof. Massimo Lopes laboratory.

**Transfections**. MUS81 siRNA was transfected at final concentration of 10 nM using Lullaby 48 h before to perform experiments. ZRANB3 siRNA was used at 40 nM using Lullaby and experiments were performed after 60 h of transfection. pCMV-tag2B (pFLAG) empty vector, pCMV-tag2B-RAD52 (pFLAG-RAD52) and pMSC-FLAG-ZRANB3 were transfected in MRC5 WT and/or MRC5 shRAD52 cell lines using Neon transfection system 48 h prior to perform experiments.

**Chemicals**. HU was added to culture medium at 2 mM from stock solutions 200 mM prepared in Phosphate-buffer saline solution (PBS) to induce DNA replication arrest or DNA damage. The B02 compound (Selleck), an inhibitor of RAD51 activity, was used at 27 μM. CldU (Sigma-Aldrich) was dissolved in sterile water as a 200 mM stock solution and used at 50 μM. IdU (Sigma-Aldrich) was dissolved in sterile DMEM as a stock solution 2.5 mM and stored at −20 °C. RAD52 inhibitor, EGC (Sigma-Aldrich) was dissolved in DMSO at 100 mM, and stock solution was stored at −80° and was used at 50 μM. Mirin, the inhibitor of MRE11 exonuclease activity (Calbiochem), was used at 50 μM. S1 nuclease (Invitrogen cat # 18001016) was diluted 1/100 in S1 buffer and 10 μl aliquots was stocked at 20° and was used at 20 U/ml.

**Western blot analysis**. Western blots were performed using standard methods. Blots were incubated with primary antibodies against: anti-MUS81 (Santa Cruz Biotechnology, 1:2000), Lamin B1 (Abcam, 1:10,000), anti-RAD52 (Santa Cruz Biotechnology, 1:500), anti-GAPDH (Millipore, 1:5000), anti-RAD51 (Bioss 1:1000), anti-SMARCAL1 (Bethyl 1:1000), anti-RPA32 (Santa Cruz Biotechnology 1:1000) and anti-ZRANB3 (Proteintech 1:1000). After incubations with horseradish peroxidase-linked secondary antibodies (Jackson ImmunoResearch, 1:30,000), the blots were developed using the chemiluminescence detection kit ECL-Plus (Amersham) according to the manufacturer's instructions. Quantification was performed on blot acquired by ChemiDoc XRS+ (Bio-Rad) using Image Lab software, and values shown on the graphs represent a normalisation of the protein content evaluated through Lamin B1 or GAPDH immunoblotting. A complete list of antibodies and dilutions is provided in Supplementary Table 2 and as Souce data file.

**Neutral comet assay**. DNA breakage induction was evaluated by Comet assay (single-cell gel electrophoresis) in non-denaturing conditions. Briefly, dust-free frosted-end microscope slides were kept in methanol overnight to remove fatty residues. Slides were then dipped into molten low melting point (LMP) agarose at 0.5% and left to dry. Cell pellets were resuspended in PBS and kept on ice to inhibit DNA repair. Cell suspensions were rapidly mixed with LMP agarose at 0.5% kept at 37 °C and an aliquot was pipetted onto agarose-covered surface of the slide. Agarose-embedded cells were lysed by submerging slides in lysis solution (30 mM EDTA, 0.1% sodium dodecyl sulfate (SDS)) and incubated at 4 °C, 1 h in the dark. After lysis, slides were washed in Tris Borate EDTA (TBE) 1X running buffer (Tris 90 mM; boric acid 90 mM; EDTA 4 mM) for 1 min. Electrophoresis was performed for 20 min in TBE 1X buffer at 1 V/cm. Slides were subsequently washed in distilled H$_2$O and finally dehydrated in ice-cold methanol. Nuclei were stained with GelRed (1:1000) and visualised with a fluorescence microscope (Zeiss), using a 60X objective, connected to a CCD camera for image acquisition. At least 300 comets per cell line were analysed using CometAssay IV software (Perceptive instruments) and data from tail moments processed using Prism software. Apoptotic cells (smaller Comet head and extremely larger Comet tail) were excluded from the analysis to avoid artificial enhancement of the tail moment.

**Immunofluorescence**. Cells were grown on 35-mm coverslips and harvested at the indicated times after treatments. For immunofluorescence (IF,) after further washing with PBS, cells were pre-extracted with 0.5% Triton X-100 and fixed with 3% para-formaldehyde (PFA)/2% sucrose at room temperature (RT) for 10 min. For RAD51, IF cells were fixed in 4% PFA/PBS and subsequently with cold methanol for 20 min. Cells were permeabilized with 0.5% Triton X-100. After blocking in 3% bovine serum albumine (BSA) for 15 min, staining was performed with mouse polyclonal anti-RPA32 (Santa Cruz, 1:300), rabbit

monoclonal anti-RAD51 (Bioss, 1:100), anti-SMARCAL1 (Abcam, 1:100) diluted in a 1% BSA/0.1% saponin in PBS solution, for 1 h at 37 °C in a humidifier chamber. Nocodazole-treated cells were blocked and fix with PTEMF (0.2% Triton X-100, 20 mM PIPES pH 6.8, 1 mM MgCl2, 10 mM EGTA and 4% formaldehyde) buffer. After blocking, coverslips were incubated for 1 h at RT with the indicated antibodies. After extensive washing with PBS, species-specific fluorophore-conjugated antibodies (Invitrogen) were applied for 1 h at RT followed by counterstaining with 0.5 mg/ml 4,6-diamidino-2-phenylindole (DAPI). Secondary antibodies were used at 1:200 dilution. A complete list of antibodies and dilutions is provided in Supplementary Table 2 and Souce data file.

**Evaluation of mitotic defects and UFBs**. Anaphase bridge and mitotic catastrophe was performed through immunofluorescence on DAPI-stained nuclei. At the end of treatments, cells were recovered for 18 h, washed two times in PBS, then fixed in 4% PFA in PBS in the dark for 10 min at RT. After two washes with PBS, cells were subjected to permeabilization with Triton X-100 0.4 % for 10 min then washed again with PBS. Staining with 0.5 μg/ml DAPI was carried out for 10 min at RT. Images were acquired as greyscale files using Metaview software (MDS Analytical Technologies) and processed using Adobe Photoshop CS3 (Adobe). For each time point, at least 200 nuclei were examined, and foci were scored at 40 ×.

For UFBs–immunofluorescence analyses, cells grown on coverslips were fixed with PTEMF buffer (20 mM PIPES pH 6.8, 0.2% Triton X-100, 1 mM MgCl2, 10 mM EGTA and 4% PFA) for 20 min. Cells were then blocked with 3% BSA in PBS for 30 min. Cells were incubated with primary antibodies diluted in 3% BSA–1% saponin in PBS for 1 h, washed with PBS and incubated with secondary antibodies diluted in 3% BSA–1% saponin in PBS for 1 h. The coverslips were washed twice with PBS and nuclei were stained with DAPI (1:4000, Serva). The following primary antibodies were used: BLM (sc-7790, Santa Cruz Biotechnology, 1:50), RPA32 (ab-3, Millipore, 1:100). Alexa Fluor 488 conjugated-goat anti-donkey and Alexa Fluor 594 conjugated-goat anti-rabbit secondary antibodies (Life Technologies) were used at 1:200.

**Chromatin isolation**. Cells (4 × 10 × 6 cells/ml) were resuspended in buffer A (10 mM HEPES, [pH 7.9], 10 mM KCl, 1.5 mM MgCl2, 0.34 M sucrose, 10% glycerol, 1 mM DTT, 50 mM sodium fluoride, protease inhibitors [Roche]). Triton X-100 (0.1%) was added, and the cells were incubated for 5 min on ice. Nuclei were collected in pellet by low-speed centrifugation (4 min, 1300 × g, 4 °C) and washed once in buffer A. Nuclei were then lysed in buffer B (3 mM EDTA, 0.2 mM EGTA, 1 mM DTT, protease inhibitors). Insoluble chromatin was collected by centrifugation (4 min, 1700 × g, 4 °C), washed once in buffer B + 50 mM NaCl, and centrifuged again under the same conditions. The final chromatin pellet was resuspended in 2X Laemmli buffer and sonicated for 15 s in a Tekmar CV26 sonicator using a microtip at 25% amplitude.

**Detection of ssDNA by native IdU assay**. To detect nascent ssDNA, cells were labelled for 20 min with 50 μM IdU (Sigma-Aldrich), immediately prior the end of the indicated treatments. To detect parental ssDNA, cells were labelled for 24 h with 50 μM IdU (Sigma-Aldrich), released in a fresh DMEM for 2 h, then treated as indicated. For immunofluorescence, cells were washed with PBS, permeabilized with 0.5% Triton X-100 for 10 min at 4 °C and fixed in 3% PFA/2% sucrose. Fixed cells were then incubated with mouse anti-IdU antibody (Becton Dickinson) for 1 h at 37 °C in 1% BSA/PBS, followed by species-specific fluorescein-conjugated secondary antibodies (Alexa Fluor 488 Goat Anti-Mouse IgG (H + L), highly cross-adsorbed—Life Technologies). Slides were analysed with Eclipse 80i Nikon Fluorescence Microscope, equipped with a Video Confocal (ViCo) system. For each time point, at least 100 nuclei were examined by two independent investigators and foci were scored at 60×. Quantification was carried out using the ImageJ software. Only nuclei showing >10 bright foci were counted as positive. Parallel samples either incubated with the appropriate normal serum or only with the secondary antibody confirmed that the observed fluorescence pattern was not attributable to artefacts.

**In situ PLA assay for ssDNA–protein interaction**. The in situ PLA (Olink, Bioscience) was performed according to the manufacturer's instructions. For nascent ssDNA–protein interaction, cells were labelled with 100 μM IdU for 20 min before treatments. After treatment, cells were permeabilized with 0.5% Triton X-100 for 10 min at 4 °C, fixed with 3% formaldehyde/2% sucrose solution for 10 min and then blocked in 3% BSA/PBS for 15 min. After washing with PBS, cells were incubated with the two relevant primary antibodies. The primary antibodies used were as follows: rabbit monoclonal anti-RAD52 (Aviva 1:150), rabbit polyclonal anti-RAD51 (Bioss, 1:100), anti-IdU (mouse monoclonal anti-BrdU/IdU; clone b44 Becton Dickinson, 1:10) and Biotin (Invitrogen, 1:500). The negative control consisted of using only one primary antibody. Samples were incubated with secondary antibodies conjugated with PLA probes MINUS and PLUS: the PLA probe anti-mouse PLUS and anti-rabbit MINUS (OLINK Bioscience). The incubation with all antibodies was accomplished in a humidified chamber for 1 h at 37 °C. Next, the PLA probes MINUS and PLUS were ligated using two connecting oligonucleotides to produce a template for rolling-cycle amplification. After amplification, the products were hybridised with red fluorescence-labelled

oligonucleotide. Samples were mounted in Prolong Gold anti-fade reagent with DAPI (blue). Images were acquired randomly using Eclipse 80i Nikon Fluorescence Microscope, equipped with a Video Confocal (ViCo) system.

**In situ PLA assay for EdU (dsDNA)–protein interaction**. Exponential growing cells were seeded onto microscope chamber slide. The day of experiment, cells were incubated with 100 μM EdU for 10 min and treated as indicated. After treatment, cells were pre-extracted in CSK-100 buffer (100 mM NaCl, 300 mM sucrose, 3 mM MgCl₂,10 mM Pipes pH 6.8, 1 mM EGTA, 0.2% Triton X-100, 1× antiproteases) for 5 min on ice under gentle agitation and fixed with 4% PFA/PBS for 20 min at RT. Cells were permeabilized in ice-cold methanol at −20 °C for 10 s and then blocked in 3% BSA/PBS for 15 min. The primary antibodies used were as follows: rabbit monoclonal anti-RAD52 (Aviva 1:150), rabbit polyclonal anti-RAD51 (Bioss, 1:100), rabbit polyclonal anti-SMARCAL1 (Abcam, 1:100), rabbit MRE11 (Bethyl, 1:1000) and Biotin (Invitrogen, 1:500). The negative control consisted of using only one primary antibody. Samples were incubated with secondary antibodies conjugated with PLA probes MINUS and PLUS: the PLA probe anti-mouse PLUS and anti-rabbit MINUS (OLINK Bioscience). The incubation with all antibodies was accomplished in a humidified chamber for 1 h at 37 °C. Next, the PLA probes MINUS and PLUS were ligated using two connecting oligonucleotides to produce a template for rolling-cycle amplification. After amplification, the products were hybridised with red fluorescence-labelled oligonucleotide. Samples were mounted in Prolong Gold anti-fade reagent with DAPI (blue). Images were acquired randomly using Eclipse 80i Nikon Fluorescence Microscope, equipped with a Video Confocal (ViCo) system.

**DNA fibre analysis**. Cells were pulse-labelled with 25 μM CldU and then labelled with 250 μM IdU with or without treatment as reported in the experimental schemes. DNA fibres were prepared and spread out as previously described[30,38]. For immunodetection of labelled tracks, the following primary antibodies were used: rat anti-CldU/BrdU (Abcam) and mouse anti-IdU/BrdU (Becton Dickinson).

Fibre assay using S1 nuclease was performed as indicated in[60]. Briefly, cells were pulse labelled as described. At end of treatment, cells were permeabilized with CSK buffer (100 mM NaCl, 10 mM MOPS, pH 7, 3 mM MgCl₂, 300 mM sucrose, 0.5% Triton X-100) for 5–10 min, then were washed with PBS and S1 nuclease buffer (30 mM sodium acetate, 10 mM zinc acetate, 5% glycerol, 50 mM NaCl, pH 4.6) prior to add S1 nuclease for 30 min at 37 °C in a humid chamber. Cells were washed with S1 buffer then with 0.1% BSA/PBS. Cells were scraped and collected pellets were used to perform fibre spreading. Images were acquired randomly from fields with untangled fibres using Eclipse 80i Nikon Fluorescence Microscope, equipped with a Video Confocal (ViCo) system. The length of labelled tracks was measured using the Image-Pro-Plus 6.0 software. A minimum of 100 individual fibres were analysed for each experiment and each experiment was repeated two times. In dot plots, the mean of at least two independent experiments are presented.

**Chromosomal aberrations**. MRC5SV40 cells were treated with HU in combination or not with RAD52 inhibitor and/or mirin at 37 °C for 4 h and allowed to recover for additional 16 h. Cell cultures were incubated with colcemid (0.2 μg/ml) at 37 °C for 3 h until harvesting. Cells for metaphase preparations were collected and prepared as previously reported[30]. For each time point, at least 100 chromosomes were examined by two independent investigators and chromosomal damage scored at 100 ×.

**DNA substrates for in vitro studies**. The polyacrylamide gel (PAGE)-purified Cy3, Cy5 and biotin-labelled oligonucleotides (see Supplementary Table 1) were custom synthesised by Integrated DNA technology (IDT). DNA substrates mimicking immobile stalled replication forks were prepared by annealing oligos #1, #2 and #3 to make G1, oligos #2, #4 and #6 to make RF1, oligos #2, #4 and #5 to make RF2, and oligos #2, #7 and #8 to make RF3. The respective oligonucleotides were mixed together at the final concentration of 1 μM each in 10 mM Tris-HCl (pH 7.5), 50 mM NaCl and 1 mM EDTA, heated to 95 °C for 5 min and slowly cooled to room temperature.

The substrate for the fork reversal experiments containing a 30 nt lagging strand gap was prepared from the 90TOP, 90BOTCy3, 50BOT and 20TOPCy5 oligos (see Supplementary Table 1). The underlined nucleotides on the parental strands form 2-bp mismatch, which prevents spontaneous branch migration. The fork DNA containing 30 nt leading strand gap was prepared by annealing the 90TOP and 90BOTCy3 strands to the 20BOT and 50TOPCy5. To prepare the fork regression substrates, leading or lagging parental strands were annealed with their corresponding nascent strands in TEN buffer [10 mM Tris-HCL (pH 8.0), 1 mM EDTA and 150 mM NaCl], and both mixtures were first heated to 95 °C for 10 min in a thermocycler and then slowly cooled to 60 °C. Then, equal amount of leading and lagging DNA intermediates were mixed and slowly cooled to 4 °C to complete annealing of the parental–parental region. Fully annealed fork DNA substrates were purified from PAGE gel using Model 442 electro-eluter system (Bio-Rad) and stored at 4 °C until use.

**EMSAs**. In all, 20 nM of G1 or RF1 DNA were mixed with the indicated concentrations of RPA and RAD52 in 10 μl of standard reaction buffer, containing 10 mM Tris-Acetate [pH 7.5], 1 mM DTT, 0.1 μg/ml BSA, 150 mM NaCl and 5 mM magnesium acetate. The reaction mixtures were incubated at 37 °C for 5 min, mixed with 1 μl of 10× Orange-G loading dye, and the free DNA was separated from the protein–DNA complexes using 0.8% Agarose (Research Products International) gel in TBE buffer (90 mM Tris [pH 8.0], 64.6 mM boric acid and 2 mM EDTA) for 2 h at 50 V using a Mupid-EX apparatus (TAKARA). The resolved species were visualised using a Chemi-doc (Bio-Rad) by exciting and monitoring Cy5 fluorescence.

**Reaction conditions for the single-molecule assay**. In total, 20 pM of G1, RF1, RF2 or RF3 DNA substrates in T50 buffer (10 mM Tris [pH 8.0], 50 mM NaCl) were immobilised on the surface of the microscope slides (Fisher Scientific), which were coated with polyethyleneglycol (PEG) to eliminate nonspecific surface adsorption of proteins. The immobilisation was mediated by biotin–neutravidin interaction between biotinylated DNA, neutravidin (Thermo Fisher), and biotinylated polymer (PEG-MW 5,000, Nectar Therapeutics). The standard buffer contained 10 mM Tris-acetate [pH 7.5], 1 mM DTT, 150 mM NaCl, 5 mM magnesium acetate and the oxygen scavenging system consisting of 1 mg/ml glucose oxidase (Sigma), 0.4% (w/v) D-glucose (Sigma), 0.04 mg/ml catalase (EMD Biosciences) and 1 mM 6-hydroxy-2,5,7,8-tetramethyl-chromane-2-carboxylic acid (Trolox) (Sigma-Aldrich). After DNA tethering on surface, indicated amount of RPA and/or RAD52 were added and incubated for 5 min at 25 °C in the standard buffer before starting the recording.

**Single-molecule data acquisition and data analysis**. Prism type TIRFM was used to excite fluorophores present on the DNA molecules. Cy3 fluorophores were excited by a DPSS laser (532 nm, 75 mW, Coherent), while the Cy5 fluorophores were excited via FRET from Cy3. The fluorescence signals originated from the Cy3 and Cy5 dyes were collected by a water immersion 60×objective (Olympus), separated by a 630 nm dichroic mirror, passed through a Cy3/Cy5 dual band-notch filter (Semrock, FF01-577/690) in the emission optical path. Images were further filtered by using a Chroma ET605/70 m filter (for Cy3 emission) and a Chroma ET700/75 m filter (for Cy5 emission) inside the dual-view system (DV2; Photometrics) and detected by the EMCCD camera (Andor) with a time resolution of 100 ms. Single-molecule trajectories were extracted from the recorded video file by IDL software. Fluorescence trajectories were analysed using customised MATLAB (The MathWorks, Inc.), scripts (available upon request from the Spies lab). To generate the FRET efficiency histograms, 20 movies with a duration of approximately 60 s were recorded in different regions of the TIRFM of slide chamber. FRET values were collected from at least 5000 molecules for each condition. FRET efficiency histograms were plotted using ORIGIN and GraphPad Prism software and fit to multiple Gaussian peaks. A zero FRET peak (3–15% of total population) representing molecules with the photo-bleached FRET acceptor (Cy5) was subtracted from the histograms. Direct excitation of the Cy5 dye using 640 nm laser was used to distinguish between the molecules displaying low FRET and molecules with the photo-bleached Cy5.

**Purification of human SMARCAL1**. Mammalian expression vector pcDNA3.1 ⁺/C-(K)-DYK containing human SMCARCAL1 cDNA (OHu16376) was purchased from Genscript (Piscataway, NJ). HEK293T cells were grown in the high glucose DMEM (Gibco) supplemented with 10% foetal bovine serum (Atlanta Biologicals), 1 mM sodium pyruvate, 1% penicillin and 1% streptomycin at 37 °C in the presence of 5% CO₂. Cells were transiently transfected for 48 h using lipofectamine 2000 (Invitrogen). Cells were washed with phosphate-buffered saline (Gibco), then resuspended in the ice-cold lysis buffer [50 mM HEPES (pH 7.5), 150 mM NaCl, 1 mM EDTA, 10% glycerol and 1 mM phenylmethanesulphonyl fluoride] and incubated at 4 °C for 30 min. After centrifugation, clarified cell lysate was mixed with M2 anti-FLAG agarose beads (Sigma-Aldrich) and then incubated at 4 °C for 2 h. Beads were then washed with lysis buffer, resuspended in the elution buffer [lysis buffer with 150 μg/ml of 3 × FLAG peptides (Sigma-Aldrich)] and incubated at 4 °C for 30 min. Eluted protein was immediately divided into small aliquots and preserved at −80 °C.

**FRET- and gel-based fork regression assay**. FRET-based analyses of DNA fork regression by SMARCAL1 in the presence of RPA and RAD52 were carried out using a Cary Eclipse fluorescence spectrophotometer (Agilent Technologies) at 30 °C in buffer [20 mM HEPES (pH 7.5), 100 mM NaCl, 5 mM MgCl₂, 100 μg/ml BSA, 2 mM ATP and 2 mM DTT]. Measurements began with buffer only (baseline) followed by addition of the 3 nM respective DNA substrate dually labelled with Cy3 and Cy5 fluorophores. Following Cy3 excitation at 530 nm, the emission of the acceptor Cy5 and donor Cy3 fluorophores were monitored simultaneously at 660 nm and 565 nm, respectively. After pre-incubation with 15 nM RPA and/or 100 nM RAD52 for 5 min, fork regression reaction was initiated upon addition of

the 1 nM SMARCAL1. The FRET signal was calculated from the Cy3 and Cy5 fluorescence at each data point as

$$\text{FRET} = \frac{I_{Cy5} \times 4.2}{I_{Cy3} \times 1.7 + I_{Cy5} \times 4.2},$$

where $I_{Cy3}$ and $I_{Cy5}$, are background corrected intensities of the two dyes. For each experiment, the FRET vs. time progress curves were plotted using GraphPad Prism 7.0 and fitted to single-exponential decay equations. The rates of the fork reversal in units of nM DNA per minute per nM SMARCAL1 were calculated as $v = \frac{k \times span}{\langle span (SMARCAL1) \rangle} \times 3\,\text{nM} \times 60$, where $k$ is the exponential decay constant, span is the FRET change between the substrate and the products, and $\langle span (SMARCAL1) \rangle$ is the average span for the reaction containing SMARCAL1 only.

For the gel-based fork regression assays, 20 nM of DNA substrates were incubated with or without 100 nM of RPA and where indicated, the indicated amounts of RAD52 at 30 °C for 5 min. Next, indicated amounts of SMARCAL1 was added in reaction buffer for 15 min at 30 °C. In all, 10 µl of each reaction was subsequently quenched and deproteinated by adding 1.5 µl of STOP solution [0.6 % SDS, 200 mM EDTA, 30% glycerol and 0.25 % Orange-G (w/v)] and further incubated for 15 min at 30 °C. The reaction products were separated by electrophoresis on the 8% (29:1) native polyacrylamide gel, visualised and quantified using the ChemiDoc MP imaging system (Bio-Rad).

**Statistical analysis.** All the data are presented as means of at least two pooled independent experiments. Statistical comparisons of wild-type RAD52-inhibited or shRAD52 cells to their relevant control were performed by one-sided analysis of variance (ANOVA) (Comet assays and restarting forks), Student's $t$-test (chromosomal damage) or Mann–Whitney test (ssDNA, PLA, DNA replication track length and other experiments) using the built-in tools in Prism 7 (GraphPad Inc.). $P < 0.05$ was considered as significant. Statistical significance was always denoted as follow: ns = not significant; *$P < 0.05$; **$P < 0.01$; ***$P < 0.001$; ***$P < 0.0001$. Any specific statistical analysis is reported in the relevant legend.

**Reporting summary.** Further information on experimental design is available in the Nature Research Reporting Summary linked to this article.

## Data availability

The authors declare that all relevant data supporting the findings of this study are available within the article and its supplementary information files and all data are available from the corresponding author upon reasonable request. Source data are provided as a Source Data file.

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

## Acknowledgements
We are grateful to Prof. Massimo Lopes (IMCR, University of Zurich) for providing the siZRANB3 oligos, the RNAi-resistant ZRANB3 expression plasmid and for scientific discussion. We thank Dr. Filomena Mazzei for critical reading of the manuscript. This work was supported by Associazione Italiana per la Ricerca sul Cancro (AIRC) to P.P. (IG17383; IG21428) and to A.F. (IG15410), and by National Institutes of Health to M.S. (NIH R01GM108617 and NIH P30 CA086862).

## Author contributions
E.M. performed the analysis of RAD52 fork recruitment in vivo and most of the functional analysis of the RAD52-dependent fork protection. G.M.P. performed the analysis of mitotic abnormalities and the detection of UFBs. M.H. performed single-molecule experiments and RAD52 and SMARCAL1 in vitro assays. V.M. performed the analysis of chromosomal damage. F.A.A. generated and characterised the inducible shSMARCAL1 cells and performed neutral Comet assay in these cells. E.M., M.H., G.M.P., F.A.A. and V.M. analysed data and contributed to designing the experiments and writing the paper. M.S., A.F. and P.P designed experiments, analysed data and wrote the paper. All authors approved the paper.

## Additional information

**Competing interests:** The authors declare no competing interests.

