## [Peer Review File · Nature Communications]

Reviewers' comments:

Reviewer #1 (Remarks to the Author):

RAD52 in yeast is the major mediator of RAD51 loading. However, in humans, BRCA2 serves this role. hRAD52's function was poorly studied until a few years ago, when Powell's lab showed that hRAD52 knockdown was synthetically lethal with BRCA2, suggesting RAD52 played a backup role. Still, the role of RAD52 has been studied primarily with respect to assays for DSB repair and at replication forks that have collapsed. In this new study, Pichierri et al follow up on their recent observation reported in eLife suggesting that RAD52 plays a role at stalled (as opposed to collapsed) replication forks. Here they perform extensive cell biological experiments supporting a model in which RAD52 provides an apparently alternative fork protection mechanism (i.e. from Mre11-dependent nascent DNA degradation) to the well studied BRCA2 pathway. Evidence that nascent DNA is degraded in an Mre11-dependent reaction when RAD52 is either knocked down or inhibited by a small molecule that interferes with its binding to ssDNA is solid. This degradation is shown to occur in S phase and RAD52 is shown to bind to both nascent and parental ssDNA at stalled forks by PLA experiments. The rest of the work is dedicated to assays attempting to explain how RAD52 minimizes stalled fork degradation. The evidence is consistent with a role for RAD52 in preventing excessive replication fork reversal, and thus degradation but a number of unexpected and unexplained results listed below prevent a firm conclusion. A strength of this work is the inclusion of biochemical experiments that provide a rational mechanism for the complex set of biological observations presented. This represents a novel contribution, a new, BRCA2-independent fork protection pathway, and the provides an extensive basis of observations for more detailed investigation of the other players in this pathway.

They first demonstrate that ssDNA accumulates in an Mre11-dependent fashion after 2h HU in the absence of RAD52 or in the presence of a RAD52 inhibitor they described previously, showing by definition that human RAD52 is involved in fork protection. Since RAD51 inhibition by a RAD51i before HU has no increase in ssDNA, they propose RAD52's role is not as a mediator of RAD51 recruitment. They also show that the role is independent of RAD52/MUS81, since MUS81 depletion does not lead to the same de-protection. Since this figure is the foundation of the whole paper, they need a control for the RAD52 knockdown showing that add back of a complementing RAD52 gene, restores fork stability in HU (Fig. 1).

Since BRCA2 recruits RAD51, they look to see if RAD52, as in yeast, acts in the same way. They show both by PLA and by different labeling protocols that RAD52 binds nascent EdU pulse-labeled dsDNA as well as both nascent and parental ssDNA after HU stalling. (Is the legend to Fig. S1 correct describing EdU as ssDNA? Does the antibody only recognize ssDNA?) RAD52 recruitment, in contradistinction to BRCA2, is not necessary for RAD51 recruitment. In fact, later they show more Rad51 is recruited in the absence of RAD52 than in its presence. Comments: One puzzling result is that at 6h after HU treatment in RAD52 depleted cells, there is a reduction instead of an increase in nascent ssDNA accumulation. What could explain this? There are also no DSBs compared to what is observed in the presence of RAD52. What is the significance of this? The EdU-PLA protocol in Methods seems incomplete.

RAD52 also inhibits RAD51 recruitment and SMARCAL1 recruitment, predicting a reduction in reversed forks to a level consistent with replication fork restart. Though this reduction in reversed forks is not directly demonstrated, the demonstration of reduction in RUVA binding (Fig. 2) is a good attempt.

It is puzzling that there is no shortening of nascent DNA in the absence of RAD52 although nascent ssDNA accumulates in an Mre11 nuclease-dependent reaction. An explanation should be inserted here. Where does the ss nascent DNA come from? Fig. 1D and S2. Is it just because only one strand is degraded, so the technique can't detect degradation? (This is rationalized later, but should be explained a bit here for the reader.)

A more puzzling result, in view of their model, is that deletion of SMARCAL1 does not decrease nascent ssDNA after RAD52 inhibition. This is especially confusing because ZRANB3 depletion reduces, as expected, ssDNA accumulation. Does Fig. 3C suggest that SMARCAL1 is playing a role in fork restoration after reversal? The reduction in ssDNA in RAD52i cells after ZRANB3 depletion they observe is less ambiguous evidence for the need for fork reversal to observe nascent ssDNA accumulation and recruitment of RAD52. (The ZRANB3 figure should be in Fig. 3 and not supplemental figures) because it is the best support for the claim that Mre11 degradation in RAD52i occurs downstream of fork reversal. They should have an add-back control for ZRANB3.

Olaparib reduces recruitment of RAD52. Therefore, they conclude that fork reversal is required for RAD52 recruitment to ssDNA. The olaparib experiment could also be explained if, as pointed out by others, olaparib prevented recruitment of Mre11 and therefore the amount of nascent ssDNA for RAD52 to bind to is lower. What is the evidence olaparib inhibits fork reversal in this case rather than Mre11-dependent degradation itself? Please clarify.

The FRET experiments seem straightforward and well done.

The fork restart experiment also seems straightforward.

Title of Fig. 6 should be, I think, Loss of RAD52... and not Loss of RAD51 as written.

Reviewer #2 (Remarks to the Author):

In this manuscript, Malacaria et al investigate the role of RAD52 at stalled replication forks. The authors find that RAD52 inhibition leads to the formation of nascent ssDNA resulting from stalled fork degradation. In addition, the authors use FRET to show that RAD52 induces a compact conformation of replication fork structures in vitro. Based on these and other findings, the authors propose a novel model in which RAD52 prevents excessive fork reversal, thus contributing to maintaining genomic stability. These findings provide new insights into the mechanisms that regulate fork stability and could be of great interest to the DNA repair community. We suggest to address the points indicated below to strengthen the manuscript.

Major Concerns

1. Figure 1D. The authors show that RAD52 inhibition causes MRE11-dependent nascent ssDNA accumulation after HU treatment. However, the authors report that MRE11 inhibition in RAD52-deficient cells reduces, rather than increases, the length of IdU tracts in DNA fiber assays (Fig. S3B). How do the authors explain the discrepancy between these findings?

2. Figure 5. The authors show that RAD52 inhibition leads to an increase of SMARCAL1 association to nascent DNA after a 2 hr HU treatment. Given the role of ZRANB3 in promoting nascent ssDNA accumulation in RAD52-inhibited cells (Fig. S7), have the authors tested whether the recruitment of ZRANB3 to nascent DNA is also altered by RAD52 inhibition?

3. Figure 8. In their model, the authors propose that RAD52 might limit fork remodeling by changing the conformation of stalled forks. To test this hypothesis, it would be helpful to determine whether RAD52 inhibits the binding of SMARCAL1 or ZRANB3 to the fork-like substrates used in Fig. 4. If this is not the case, the authors could also examine whether RAD52 limits the remodeling of synthetic fork substrates by SMARCAL1 or ZRANB3 (Betous et al, Genes Dev, 2012; Betous et al, Cell Rep, 2013). Alternatively, the authors could determine whether RAD52 inhibition (or overexpression), combined or not to mirin treatment, alters the number of reversed forks, as detected by electron microscopy.

Minor Concerns

1. Figure 1B. The timing of RAD52 inhibition to detect nascent ssDNA differs between untreated (60 min) and HU-treated conditions (2-6 h). What is the effect of a 2-6 h RAD52 inhibitor treatment on nascent ssDNA without HU?

2. In line 111 the authors write "we labelled replicating cells with EdU 15 minutes before adding IdU". Was the EdU labeling conducted for 15 min or 10 min, as indicated in Fig. S1A?

Reviewer #3 (Remarks to the Author):

In this work, the authors report a previously uncharacterized function of Rad52, the protection of stalled replication fork from excessive replication fork reversal by fork reversal enzymes such as SMARCAL1 and eventual degradation by MRE11. The conclusion is based mostly on extensive cell studies. To support the conclusion further, they also performed single-molecule FRET experiments, and showed that Rad52 binding on replication fork induces compaction of the replication forks. I found the paper is interesting, but single-molecule part can be improved. For example, single-molecule FRET has been used to monitor the replication fork reversal of various enzymes such as WRN and Rad5. The author can use the same assay to directly prove their conclusion. Overall, the work reports an interesting discovery, and I recommend a publication of the paper on Nature Communications on condition that single-molecule part is improved.

Point-by-point reply to reviewers

Reviewer #1:

We really appreciated the highly laudatory comment on the significance, novelty and relevance of our work made by reviewer #1. We also appreciated the specific comments aimed to improve the strength of our affirmations; they were all helpful and allowed us to improve the manuscript by responding to all of them with additional experiments, which are now included in the revised manuscript. We hope that our effort is sufficient to convince the reviewer.

Below, the reviewer will find our specific replies to each comment/suggestion

They first demonstrate that ssDNA accumulates in an Mre11-dependent fashion after 2h HU in the absence of RAD52 or in the presence of a RAD52 inhibitor they described previously, showing by definition that human RAD52 is involved in fork protection. Since RAD51 inhibition by a RAD51i before HU has no increase in ssDNA, they propose RAD52's role is not as a mediator of RAD51 recruitment. They also show that the role is independent of RAD52/MUS81, since MUS81 depletion does not lead to the same de-protection. Since this figure is the foundation of the whole paper, they need a control for the RAD52 knockdown showing that add back of a complementing RAD52 gene, restores fork stability in HU (Fig. 1).

Reviewer made an excellent point. We have now included additional data showing the add-back of RAD52 in our shRAD52 cell line is able to revert the accumulation of nascent ssDNA. These data are now presented in Figure 1 as panels C and D.

Since BRCA2 recruits RAD51, they look to see if RAD52, as in yeast, acts in the same way. They show both by PLA and by different labeling protocols that RAD52 binds nascent EdU pulse-labeled dsDNA as well as both nascent and parental ssDNA after HU stalling. (Is the legend to Fig. S1 correct describing EdU as ssDNA? Does the antibody only recognize ssDNA?) RAD52 recruitment, in contradistinction to BRCA2, is not necessary for RAD51 recruitment. In fact, later they show more Rad51 is recruited in the absence of RAD52 than in its presence. Comments: One puzzling result is that at 6h after HU treatment in RAD52 depleted cells, there is a reduction instead of an increase in nascent ssDNA accumulation. What could explain this? There are also no DSBs compared to what is observed in the presence of RAD52. What is the significance of this? The EdU-PLA protocol in Methods seems incomplete.

The click-it reaction used to detect EdU incorporation is not specific for ssDNA and has been used to detect either replication sites in interphase nuclei or to detect fork-recruitment at single-cell level by coupling the Click-it reaction with PLA. Figure S1 refers to the dual-detection of replicating sites (EdU) and ssDNA (native IdU IF). It is possible that the structure of the legend was troublesome and generated confusion. We changed the sentence to make it clearer.

Concerning the reduction of nascent ssDNA observed at 6h in the absence of RAD52, this phenotype is only apparently at odds with the increase reported at earlier time-point of HU. Indeed, as shown in Figure 1 and Figure S2, at 6h treatment with HU starts to produce DSBs in wild-type cells. These DSBs are formed through the combined action of RAD52 and MUS81, as reported in our eLife and Plos Genetics papers and as shown in the figure below (panel A), which we have not included in the manuscript because of the already elevated character count. Thus, according to these findings, the reduction of the nascent ssDNA observed at 6h correlates with reduction in DSBs formation and their resection. Indeed, as shown in the panel B of the figure below, depletion of MUS81 also reduces nascent ssDNA in wild-type cells.

Finally, we extended the EdU-PLA method description.

RAD52 also inhibits RAD51 recruitment and SMARCAL1 recruitment, predicting a reduction in reversed forks to a level consistent with replication fork restart. Though this reduction in reversed forks is not directly demonstrated, the demonstration of reduction in RUVA binding (Fig. 2) is a good attempt.

We appreciated the reviewer's positive comment on the use of RuvA as a surrogate for direct observation of reversed forks.

It is puzzling that there is no shortening of nascent DNA in the absence of RAD52 although nascent ssDNA accumulates in an Mre11 nuclease-dependent reaction. An explanation should be inserted here. Where does the ss nascent DNA come from? Fig. 1D and S2. Is it just because only one strand is degraded, so the technique can't detect degradation? (This is rationalized later, but should be explained a bit here for the reader.)

We agree with the reviewer's comment. Our best explanation is indeed that only one strand gets extensively degraded while the other is left unaffected. We now give a better rationale for the observed result and we have added another DNA fiber experiment performed after S1 treatment of cells before fiber spreading. The rationale is that S1 nuclease degrades ssDNA regions and thus it should artificially reduce the length of nascent strand if it is in a ssDNA form, at least in part. The result of this experiment is now presented in Fig. S4 and clearly shows that S1 pre-treatment before fiber spreading and immunodetection leads to a significant shortening of the replication tracks of cells inhibited of RAD52. Consistent with our ssDNA data, Mirin treatment restores the track length. Having added this important experiment, we removed the original Fig. S3C showing the IdU/CldU ratio as it provided no additional information to the DNA fiber analyses.

A more puzzling result, in view of their model, is that deletion of SMARCAL1 does not decrease nascent ssDNA after RAD52 inhibition. This is especially confusing because ZRANB3 depletion reduces, as expected, ssDNA accumulation. Does Fig. 3C suggest that SMARCAL1 is playing a role in fork restoration after reversal? The reduction in ssDNA in RAD52i cells after ZRANB3 depletion they observe is less ambiguous evidence for the need for fork reversal to observe nascent ssDNA accumulation and recruitment of RAD52. (The ZRANB3 figure should be in Fig. 3 and not supplemental figures) because it is the best support for the claim that Mre11 degradation in RAD52i occurs downstream of fork reversal. They should have an add-back control for ZRANB3.

We agree with the reviewer in defining the critical role of SMARCAL1 in fork recovery. Indeed, the reason behind our inability to detect a reduction of ssDNA in cells inhibited of RAD52 after SMARCAL1 depletion is that the concomitant abrogation of both activities results in many DSBs (Fig. S8) that confound analysis of ssDNA, which increases because of enhanced DSBs resection and its takes over.

We now added new data (Fig. S8E,F) showing that depletion of ZRANB3 does not trigger DSBs formation in RAD52i cells, which complements our ability to detect a reduced formation of ssDNA after ZRANB3 depletion in cells treated with the RAD52 inhibitor.

We have moved the ssDNA data obtained depleting RAD52i cells of ZRANB3 from supplementary material to primary figures and we have included in Fig. S7C,D the add-back experiments by re-introducing ZRANB3 into ZRANB3 siRNA-treated cells.

Olaparib reduces recruitment of RAD52. Therefore, they conclude that fork reversal is required for RAD52 recruitment to ssDNA. The olaparib experiment could also be explained if, as pointed out by others, olaparib prevented recruitment of Mre11 and therefore the amount of nascent ssDNA for RAD52 to bind to is lower. What is the evidence olaparib inhibits fork reversal in this case rather than Mre11-dependent degradation itself? Please clarify.

Previous study demonstrated that Olaparib treatment before replication stress reduces the formation of reversed forks as evaluated by EM. Similarly, as pointed out by the reviewer, Olaparib treatment can interfere with MRE11 recruitment.

However, we used Olaparib in wild-type cells in order to evaluate RAD52 recruitment to ssDNA after HU, a condition that does not result in MRE11-dependent ssDNA formation, as shown in Fig.1. Moreover, Olaparib treatment does not affect ssDNA formation in wild-type cells (Old Fig. 3A and S6), that is under conditions in which we perform RAD52-ssDNA PLA.

That said, we consider unlikely that the effect of Olaparib on RAD52 recruitment at ssDNA is mediated by inhibition of MRE11-dependent degradation and ssDNA formation since this does not, or does barely, occur in wild-type cells on short HU treatments.

The FRET experiments seem straightforward and well done.

We appreciate the reviewer's comment.

The fork restart experiment also seems straightforward.

We appreciate it.

Title of Fig. 6 should be, I think, Loss of RAD52... and not Loss of RAD51 as written.

We have amended the title.

Reviewer #2:

We would thank the reviewer for the appreciation of our work. We found the comments and suggestions listed in the reviewing's report insightful and useful. We tried to address all the points raised by the reviewer by including additional experiments that, is what we hope, should have contributed to strengthen our conclusions.

Below, we provide our point-by-point reply to the specific comments.

Major Concerns

1. *Figure 1D. The authors show that RAD52 inhibition causes MRE11-dependent nascent ssDNA accumulation after HU treatment. However, the authors report that MRE11 inhibition in RAD52-*

deficient cells reduces, rather than increases, the length of IdU tracts in DNA fiber assays (Fig. S3B). How do the authors explain the discrepancy between these findings?

Our hypothesis is that the loss of RAD52 function triggers something like a “unidirectional” MRE11-dependent degradation of nascent strand, which involves trimming of just one of the two nascent strands. This phenomenon could account for our inability to detect any shortening of the replication tracks on HU in RAD52i cells. As shown in Figure 8, Mirin treatment in RAD52i cells reduces fork restart, suggesting that MRE11-dependent degradation also participates to fork restart. It might be possible that the reduction in the IdU tract length is correlated with this MRE11 role in the absence of RAD52, although it is unlike that fork restart occurs on HU. We do not have a definitive answer, unfortunately.

However, we now included additional DNA fiber experiments performed by treating cells with the ssDNA nuclease S1 before spreading DNA fibers. The results of this experiment are presented in Fig. S4 of the revised version of the manuscript. They clearly show that S1 treatment results in reduced IdU tracts length in RAD52i cells but not in mock-inhibited cells. This supports our claim of the degradation by MRE11 of just one nascent strand leaving the other unaffected and able to support IdU immunodetection after denaturation of DNA fibres.

2. Figure 5. The authors show that RAD52 inhibition leads to an increase of SMARCAL1 association to nascent DNA after a 2 hr HU treatment. Given the role of ZRANB3 in promoting nascent ssDNA accumulation in RAD52-inhibited cells (Fig. S7), have the authors tested whether the recruitment of ZRANB3 to nascent DNA is also altered by RAD52 inhibition?

We thank the reviewer for the useful critique and point. We now included additional EdU-PLA data showing fork recruitment of ZRANB3 in RAD52i cells (Fig. S12). The new data show that inhibition of RAD52 increases the fork-recruitment of ZRANB3 as it increases that of SMARCAL1. It should be noted that we had to perform ZRANB3 EdU-PLA experiments after transfection of cells with an HA-tagged ZRANB3. Indeed, available anti-ZRANB3 antibodies do not perform well in PLA making it impossible to detect fork recruitment of endogenous ZRANB3, at least under our conditions.

3. Figure 8. In their model, the authors propose that RAD52 might limit fork remodeling by changing the conformation of stalled forks. To test this hypothesis, it would be helpful to determine whether RAD52 inhibits the binding of SMARCAL1 or ZRANB3 to the fork-like substrates used in Fig. 4. If this is not the case, the authors could also examine whether RAD52 limits the remodeling of synthetic fork substrates by SMARCAL1 or ZRANB3 (Betous et al, Genes Dev, 2012; Betous et al, Cell Rep, 2013). Alternatively, the authors could determine whether RAD52 inhibition (or overexpression), combined or not to mirin treatment, alters the number of reversed forks, as detected by electron microscopy.

We now included gel- and FRET-based experiments which monitor SMARCAL1-mediated synthetic DNA fork reversal in the presence and absence of RPA (Fig. 5 and S10). In these experiments, we demonstrate that RAD52 binding to the fork does indeed inhibit fork remodelling by SMARCAL1, thus supporting our model.

Minor Concerns

1. Figure 1B. The timing of RAD52 inhibition to detect nascent ssDNA differs between untreated (60 min) and HU-treated conditions (2-6 h). What is the effect of a 2-6 h RAD52 inhibitor treatment on nascent ssDNA without HU?

We initially performed also longer incubation with the RAD52i and we observed a reduced efficiency in replication, contributing to mixed results. Thus, we opted for the shorter inhibition time before IdU pulse.

2. *In line 111 the authors write “we labelled replicating cells with EdU 15 minutes before adding IdU”. Was the EdU labeling conducted for 15 min or 10 min, as indicated in Fig. S1A?*

It was a typo. The correct labelling time is 10min. We have amended the sentence accordingly.

Reviewer #3:

In this work, the authors report a previously uncharacterized function of Rad52, the protection of stalled replication fork from excessive replication fork reversal by fork reversal enzymes such as SMARCAL1 and eventual degradation by MRE11. The conclusion is based mostly on extensive cell studies. To support the conclusion further, they also performed single-molecule FRET experiments, and showed that Rad52 binding on replication fork induces compaction of the replication forks. I found the paper is interesting, but single-molecule part can be improved. For example, single-molecule FRET has been used to monitor the replication fork reversal of various enzymes such as WRN and Rad5. The author can use the same assay to directly prove their conclusion. Overall, the work reports an interesting discovery, and I recommend a publication of the paper on Nature Communications on condition that single-molecule part is improved.

We thank the reviewer for the appreciation of our work. The single-molecule and FRET data are crucial in our study. Moreover, a direct *in vitro* demonstration of the RAD52 ability to inhibit a SMARCAL1-mediated fork reversal will further validate our model. Other reviewers also suggested a similar experiment. Therefore, we now included a demonstration of the replication fork reversal by SMARCAL1 and its inhibition by RAD52 as new Figures 5 and S10. SMARCAL1 has been selected because of its crucial role *in vivo* and for consistency with the cell studies. Complementary gel-based and FRET-based analyses are presented. Our new data demonstrate that RAD52 prevents SMARCAL1 from reversing synthetic fork structures both in the presence and in the absence of RPA, hence strongly supporting our claims and final model.

We hope that the reviewer may find the new data compelling and interesting.

For all reviewers:

We noticed that in the Fig.1A of the original submission the control WB was erroneously labelled as GAPDH while it referred to LAMINB1. We now fixed the error by correctly labelling the blot. We apologise.

REVIEWERS' COMMENTS:

Reviewer #1 (Remarks to the Author):

The manuscript is much improved and ready for publication except for the need for some clarification of new data. The new figure 5 shows that RAD52 inhibits SMARCAL fork reversal. The description of the experiment is very brief however. First, there is no discussion of why RPA inhibits SMARCAL1 inhibition of fork reversal, although they reference that that was observed by others previously. Some explanation should be in the text. Second, in Figures F, all experiments did have DNA didn't they? The legend leaves the DNA out of two reactions. Third, in Figure F, why is there both a dark and a light green curve for the SMARCAL +RPA and a gray and black curve for the SMARCAL alone? What are the brackets in each of the F and G figures?

Reviewer #2 (Remarks to the Author):

The authors have satisfactorily addressed the majority of the reviewer's points. The authors now include in their revised manuscript new findings that solidify their previous observations on the role of RAD52 at the replication fork. We therefore recommend this manuscript for publication.

Reviewer #3 (Remarks to the Author):

The authors addressed my comment satisfactorily, and I recommend the publication of the paper in Nature Communications.

Point-by-point reply to reviewers

Reviewer #1:

We really appreciated the highly laudatory comments of the reviewer. We added now more on the description of the experiment and on the significance of RPA-mediated inhibition if RPA is located at lagging strand gap, also referring to another original paper.

We also amended the legend of fig. 5 for clarity and to include missing information.